# KIBRA controls exosome secretion via inhibiting the proteasomal degradation of Rab27a

Lin Song[1], Shi Tang[1], Xiaolei Han[1], Ziying Jiang[1], Lingling Dong[2], Cuicui Liu[1], Xiaoyan Liang[1], Jixin Dong[3], Chengxuan Qiu[1,4], Yongxiang Wang[1] & Yifeng Du[1]

Exosomes are nanosized membrane vesicles released from cells after fusion of multivesicular bodies (MVBs) with the plasma membrane (PM) and play important roles in intercellular communication and numerous biological processes. However, the molecular mechanisms regulating exosome secretion remain poorly understood. Here we identify KIBRA as an adaptor-like protein that stabilizes Rab27a, which in turn controls exosome secretion both in vitro and in vivo. Knockdown or overexpression of KIBRA in neuronal and podocyte cell lines leads to a decrease or increase of exosome secretion, respectively, and KIBRA depletion increases MVB size and number. Comparing protein profiles between KIBRA knockout and wild-type mouse brain showed significantly decreased Rab27a, a small GTPase that regulates MVB-PM docking. Rab27a is stabilized by interacting with KIBRA, which prevents ubiquitination and degradation via the ubiquitin-proteasome pathway. In conclusion, we show that KIBRA controls exosome secretion via inhibiting the proteasomal degradation of Rab27a.

[1] Department of Neurology, Shandong Provincial Hospital affiliated to Shandong University, 250021 Jinan, Shandong, China. [2] Department of Neurology, Dongying People's Hospital, 257091 Dongying, Shandong, China. [3] Eppley Institute for Research in Cancer and Allied Diseases, University of Nebraska Medical Center, Omaha, NE 68198, USA. [4] Aging Research Center, Department of Neurobiology, Care Sciences and Society, Karolinska Institutet-Stockholm University, 17177 Stockholm, Sweden. These authors contributed equally: Lin Song, Shi Tang. Correspondence and requests for materials should be addressed to Y.W. (email: wang-yongxiang@hotmail.com) or to Y.D. (email: du-yifeng@hotmail.com)

Exosomes are nanovesicles of 30–150 nm in diameter that participate in diverse extracellular functions such as immune function, metabolic regulation, tumor metastasis, and neurodegeneration[1,2]. Exosomes develop from in-budding of early endosomes, which, in turn, forms multivesicular bodies (MVBs) that contain intraluminal vesicles (ILVs). Some MVBs then fuse with the plasma membrane (PM) to release ILVs to extracellular environment as exosomes. Alternatively, some MVBs are delivered to lysosomes where their cargo, such as proteins, is degraded and parts of degraded products are recycled[3]. Precise regulation of exosome secretion is critical for normal cell-to-cell communication.

The molecular mechanisms that directly govern exosome secretion and trafficking have been extensively studied. Recent studies have identified several essential regulators of exosome biogenesis and secretion in diverse cell types[4–7]. Endosomal sorting complexes required for transport proteins (e.g., HRS and Tsg101), lipids (e.g., ceramide), and tetraspanins (e.g., CD81 and CD9) have been demonstrated to regulate exosome secretion by regulating MVB biogenesis[6,8,9]. Some Rab GTPases (e.g., Rab11, Rab27, and Rab35) have also been shown to regulate exosome release, probably by affecting transport or docking of MVBs to the target PM[10–12]. Furthermore, soluble N-ethylmaleimide-sensitive factor-attachment protein receptor complexes are instrumental in allowing fusion of the lipid bilayers after docking of two different intracellular compartments[13,14]. However, the upstream platform for exosome regulators is not well understood.

KIBRA is predominantly expressed in the kidney and in the memory-related brain regions, where it functions as a scaffold protein in various cell processes, such as cell polarity, cell migration, and membrane trafficking[15,16]. In Drosophila, genetic studies have identified KIBRA as a regulator of the Hippo signaling pathway, which plays an important role in tumorigenesis[17–19]. In neurons, KIBRA plays a critical role in regulating α-amino-3-hydroxyl-5-methyl-4-isoxazole-propionate receptor trafficking underlying synaptic plasticity and learning[20]. It has been reported that KIBRA regulates the long-range transport of early endosomes through an interaction with dynein light chain 1, which is a component of cytoplasmic dynein[21]. In addition, KIBRA knockdown (KD) accelerates the exocytosis of the apical protein to the cell surface through inhibition of aPKC kinase activity[22,23]. However, the potential roles for KIBRA in regulating exosome or extracellular vesicle (EV) secretion have yet to be elucidated.

In the present study, our investigations reveal that the absence of KIBRA dramatically decrease the secretion of EVs and increase the size and number of intracellular MVBs both in vitro and in vivo. The expression of Rab27a, a small GTPase reported to regulate MVB docking to the PM, is significantly downregulated at the protein level but not at the mRNA level when KIBRA is depleted. In addition, protein degradation of Rab27a can be restored by the proteasome inhibitor lactacystine (Lac) but not by the lysosome inhibitor bafilomycin A1 (Baf). We further confirm the co-localization and association of KIBRA with Rab27a through immunofluorescence staining and cross-immunoprecipitation (cross-IP) experiments, respectively. In summary, we conclude that KIBRA, as an adaptor-like protein, plays an essential role stabilizing Rab27a against being degraded which, in turn, regulates the exosome secretion.

## Results

**KIBRA regulates secretion of EVs in vitro.** In mammals, KIBRA is predominantly expressed in the kidney and brain (Supplementary Fig. 1). Thus we performed in vitro experiments using a mouse hippocampal neuronal cell line (HT22) and a mouse

podocyte cell line (MPC5). KIBRA was downregulated or overexpressed in these two cell lines, and the results were confirmed by western blot and quantitative real-time polymerase chain reaction (qRT-PCR) analysis (Supplementary Fig. 2). To analyze the effects of KIBRA on the secretion of different subtypes of EVs, KIBRA-KD or -overexpressed cells were cultured in Dulbecco's modified Eagle's medium (DMEM) or RPMI-1640 medium containing 10% exosome-depleted fetal bovine serum (FBS) for 48 h, and the EVs were isolated from the conditioned media by sequential differential centrifugation. Pellets recovered by low ($2000 \times g$) or medium ($10,000 \times g$) centrifugation speed together with the ultracentrifuged pellets (small EVs classically considered to be exosomes) were washed and lysed for further analysis.

We first quantified the total amount of proteins pelleted at $2000 \times g$ (2K pellet), $10,000 \times g$ (10K pellet), and $100,000 \times g$ (small EVs) with a BCA kit. The results indicated a decrease in the 2K and 10K pellets obtained from KIBRA-KD cells compared with Ctrl-KD cells, but the differences were not statistically significant (Supplementary Fig. 3A, B). However, the total amount of protein isolated by ultracentrifugation was significantly decreased in KIBRA-KD cells compared with control cells, as shown in Fig. 1a.

To further characterize the different subtypes of EVs, widely recognized exosome markers were analyzed in 2K pellet, 10K pellet, small EVs, and whole cell lysates (WCL) by western blot. The exosome markers Alix, CD63, Tsg101, and CD9 were highly abundant not only in small EVs but also in the 2K and 10K pellets. The exosome-excluded endoplasmic reticulum protein Calnexin was hardly detectable in small EVs but was abundant in the 2K and 10K pellets as well as the WCL, indicating that exosomes in the ultracentrifuged pellets were relatively pure without contamination of other cell compartments, while the large EVs contained various parts of secreting cells. As the vesicles were isolated from equivalent numbers of cells, the intensity of the exosomal markers reflected the ability of cells to secrete EVs. Knockdown of KIBRA in HT22 cells led to a significant decrease of Alix, Tsg101, CD63, and CD9 in the ultracentrifuged pellets but not in the WCL, suggesting that the differences in secretion levels are due to impaired secretion of EVs rather than decreased intracellular levels of the EVs markers (Fig. 1b, c). However, only a modest decrease of CD9 or CD63 was observed in the 2K and 10K pellets, and the level of Alix and Tsg101 remained unchanged (Supplementary Fig. 3C–E). These results suggest that KIBRA may regulate secretion of different types of EVs, most notably, the secretion of exosomes, due to its pleiotropic effect.

Consistent with these results, the nanoparticle tracking analysis (NTA) of the ultracentrifuged pellets revealed a decrease in the number of particles secreted by KIBRA-KD cells compared with control cells, demonstrating that downregulating KIBRA did not cause a mere decrease in ultracentrifuged pellets but an actual decrease in secretion of small EVs (Fig. 1e, f). Furthermore, NTA and electron microscopic analyses revealed that the majority of vesicles pelleted by ultracentrifugation had the size similar to those generally described for exosomes (i.e., 50–150 nm). Small EVs secreted by KIBRA-KD cells were identical in size and morphology to those produced by control cells (Fig. 1d–f).

In contrast, after overexpressing KIBRA in HT22 cells via infection with KIBRA-expressing lentivirus, exosome secretion was remarkably enhanced, as indicated by both measuring the total amount of exosomal proteins and immunoblotting analysis of exosome markers (Fig. 1g–i). These data indicate that KIBRA significantly regulates the secretion of exosomes in HT22 cells without affecting the size and morphology of exosomes.

To further explore whether KIBRA-regulated EV secretion is cell specific, we used CRISPR-Cas9 gene editing system to

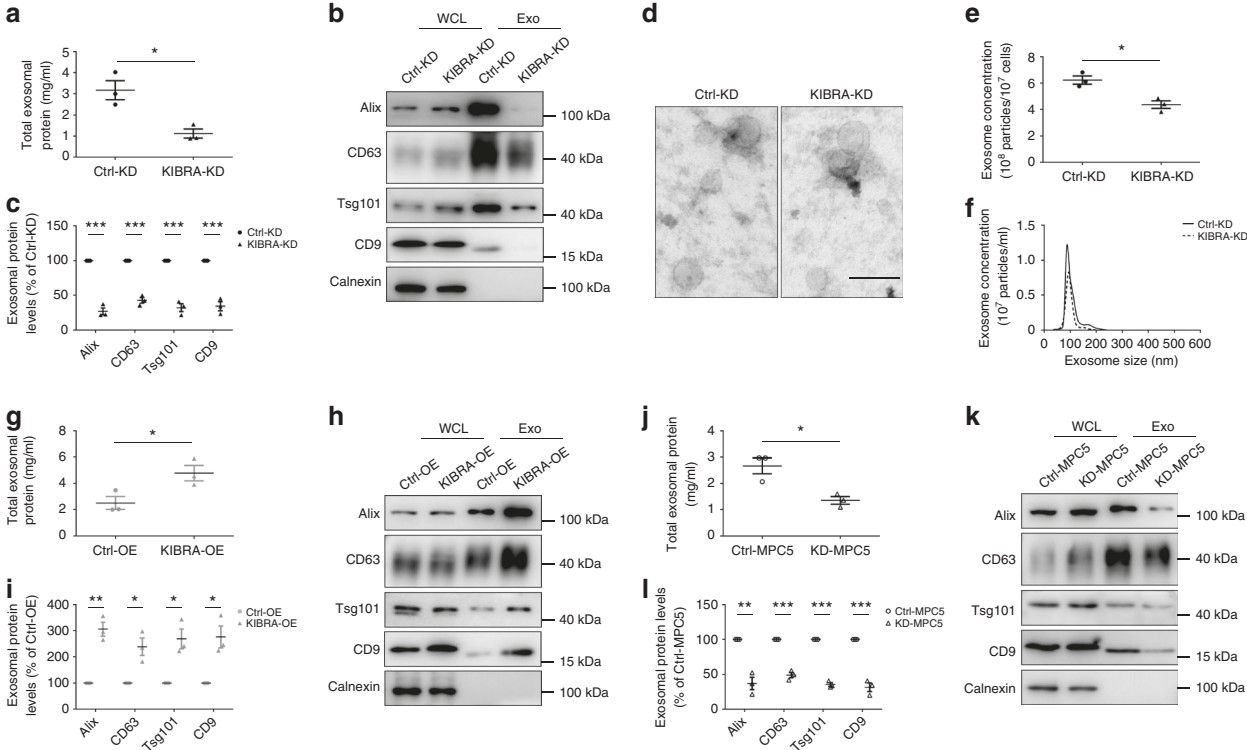

**Fig. 1** KIBRA regulates secretion of small extracellular vesicles (EVs) in vitro. **a** Concentrations of exosomal proteins in KIBRA-KD and Ctrl-KD cells. Small EVs were isolated by serial ultracentrifugation from cell culture supernatants of 20 million cells and resuspended in 30 μl lysis buffer. **b** Western blot analysis of small EVs purified by serial ultracentrifugation from cell culture supernatants from equal numbers of KIBRA-KD and Ctrl-KD cells. Whole cell lysates (WCL) and small EVs (Exo) were blotted for the exosomal markers Alix, CD63, Tsg101, and CD9 and for the endoplasmic reticulum marker Calnexin. **c** Quantification of exosomal protein levels in the small EVs obtained from KIBRA-KD and Ctrl-KD cells in three independent experiments. **d** Small EVs purified from cell culture supernatants were negatively stained and representative electron microscopic images were shown. Scale bar = 100 nm. **e** Quantification of nanoparticle tracking analysis (NTA) of three independent experiments. **f** Representative NTA traces of exosomes derived from KIBRA-KD and control cells, normalized to cell number. **g** Concentrations of exosomal proteins in KIBRA-OE and Ctrl-OE cells. Small EVs were isolated by serial ultracentrifugation from cell culture supernatants of 20 million cells and resuspended in 30 μl lysis buffer. **h** Western blot analysis of EVs purified from equal numbers of KIBRA-OE and Ctrl-OE cells. **i** Quantification of exosomal protein levels in the EVs obtained from KIBRA-OE and Ctrl-OE cells in three independent experiments. **j** Concentration of exosomal proteins in Ctrl-MPC5 and KD-MPC5 cells. Small EVs were isolated by serial ultracentrifugation from cell culture supernatants of 20 million cells and resuspended in 30 μl lysis buffer. **k** Western blot analysis of EVs purified from equal numbers of Ctrl-MPC5 and KD-MPC5 cells. **l** Quantification of exosomal protein levels in the EVs obtained from KD-MPC5 and KD-MPC5 cells in three independent experiments. All quantification results were plotted as dot plots, showing the mean ± SE of three independent experiments. $*P < 0.05$, $**P < 0.01$, $***P < 0.001$ as determined by two-tailed $t$ test

downregulate KIBRA in MPC5 cells. The results showed that the total amount of exosomal proteins as well as the levels of exosomal markers (Alix, CD63, Tsg101, and CD9) decreased significantly in MPC5-KD cells compared with control cells, which was consistent with the results observed in HT22 cells (Fig. 1j–l).

**KD of KIBRA increases the size and number of MVBs**. To gain further insight into the mechanisms for impaired exosome secretion in KIBRA-KD cells, we performed an electron microscopic analysis to investigate the number and morphology of ILVs and MVBs. Interestingly, although exosome secretion significantly decreased when KIBRA was depleted, the number of MVBs per cell and the number of ILVs per MVB dramatically increased compared with control cells, indicating that the absence of KIBRA may result in the abnormal accumulation of ILVs in MVB (Fig. 2a–d). Meanwhile, the morphology of ILVs did not show any significant change, suggesting that KIBRA may only affect the exosome secretion rather than its generation (Fig. 2a).

To further investigate the subcellular distribution of MVBs, immunofluorescence imaging was performed using CD63 as a

marker. KIBRA KD resulted in a more clustered localization of CD63, as shown by the increased size and number of CD63⁺ spots per cell (Fig. 2e, f). Accordingly, a western blot analysis suggested that the expression level of CD63 in KIBRA-KD cells was almost three times that in Ctrl-KD cells (Fig. 2i, j). However, in both KIBRA-KD cells and Ctrl-KD cells, CD63-positive MVBs appeared to be homogenously distributed throughout the cells. Similar results were obtained with another MVB marker LBPA[24] (Fig. 2g, h). As MVBs are formed by inward budding of the early endosomal membrane, the increased number of MVBs could be due to either a massive increase in the biogenesis of MVBs or inhibited transport and degradation of MVBs. We compared the expression level and subcellular distribution of the early endosome marker EEA1 in KIBRA-KD cells and control cells. Both immunofluorescence imaging and western blot analyses did not show any apparent alteration of EEA1 (Supplementary Fig. 4A, B, G, H), preliminarily ruling out the potential effect of KIBRA in MVB formation.

It has been reported that exosome secretion may be directly linked with autophagy, a degradation pathway of aggregated proteins that supplies nutrients during starvation[25,26]. Once autophagy is initiated, the cytoplasmic cargo is sequestered

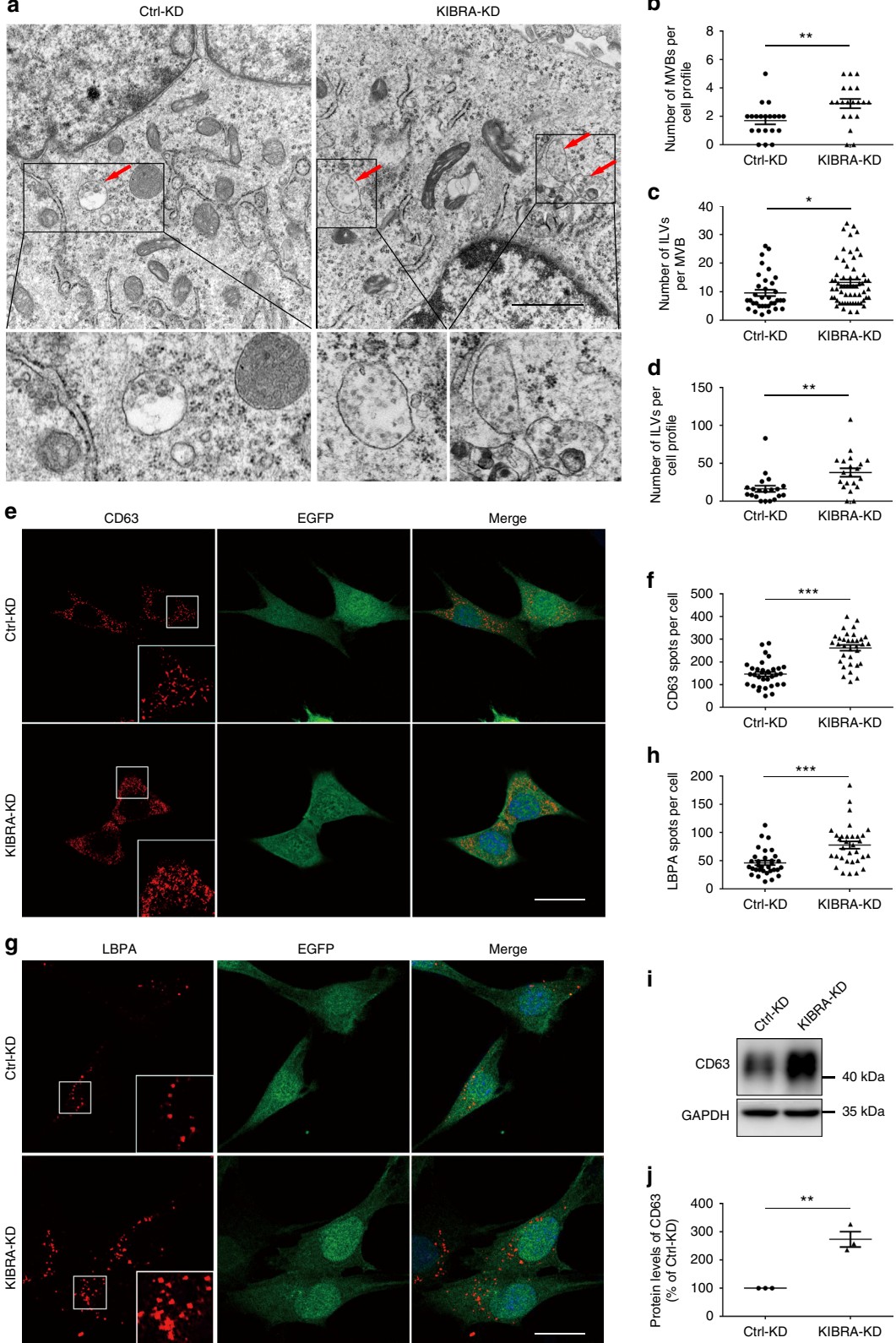

within double-membrane vesicles termed autophagosomes, which can fuse with MVBs to form amphisomes, or directly fuse with lysosomes for cargo degradation[27]. However, both western blot and immunofluorescence assays did not show any apparent alteration in the levels of LC3B (a marker of autophagosomes) or Lamp2 (a marker of lysosomes) (Supplementary Fig. 4C–H).

Taken together, these data support the view that KIBRA does not control the biogenesis of MVBs or the autophagy process but mainly affects later stages in the MVB pathway.

**KIBRA regulates exosome secretion in vivo.** To define the role of KIBRA in exosome secretion in a more physiological context,

**Fig. 2** Increases in size and number of multivesicular bodies (MVBs) in KIBRA-KD cells. **a** Representative electron microscopic images of KIBRA-KD and control cells. Scale bar = 1 μm. Zooms (lower panels) are ×4.8. Red arrows indicate MVBs containing typical intraluminal vesicles (ILVs). **b** The number of MVBs per cell profile. **c** The number of ILVs per MVB. **d** The number of ILVs per cell profile. ILVs and MVBs in 20 profiles of different cells were counted in a blind manner and only MVBs containing typical ILVs were counted. **e**, **g** Confocal microscopic analysis of KIBRA-KD and Ctrl-KD cells stained with anti-CD63 (**e**, red) and anti-LBPA (**g**, red) antibodies. KIBRA-knockdown cells were generated using CRISPR-Cas9 gene editing system, and cells transfected with LV-sgRNA (KIBRA-KD cells) or control vectors (Ctrl-KD cells) could stably express EGFP (green). Scale bar = 20 μm. Zooms were ×5.6. **f**, **h** Quantification of the number of CD63[+] (**f**) and LBPA[+] (**h**) particles per cell. The dot plot represents the number of CD63[+] or LBPA[+] particles from individual cells (n > 30). The CD63[+] and LBPA[+] particles were counted in a blind manner. **i**, **j** Western blot analysis (**i**) and quantification (**j**) of CD63 in KIBRA-KD and control cells. Quantification result were plotted as dot plots, showing the mean ± SE of three independent experiments. *P < 0.05, **P < 0.01, ***P < 0.001 as determined by two-tailed t test

we isolated and purified exosomes from the extracellular space of the brain and kidney in KIBRA-knockout (KO) mice and their wild-type (WT) counterparts by sucrose density gradient centrifugation. The contents of the sucrose step gradient fractions (a–g) were immunoblotted for proteins known to be either enriched (CD63 and CD9) or absent (Calnexin) in exosomes. As expected, the brain cell lysate and kidney cell lysate were enriched with Calnexin, an endoplasmic reticulum protein, whereas Calnexin was not detected in an equivalent amount of the sucrose step gradient fractions (a–g) (Fig. 3a, c). This finding indicates that the sucrose step gradient fractions were not contaminated with cellular debris. In addition, immunoblotting of the sucrose step gradient fractions (a–g) demonstrated that CD63 and CD9 were present mainly in four of the fractions (b–e) but not in the other fractions (a, f, and g) (Fig. 3a, c). To explore whether KIBRA also regulates exosome secretion in vivo, fractions b–e isolated from the extracellular space of the brain or kidney in KIBRA-KO and -WT mice were blotted for the exosomal markers CD63 and CD9. Consistent with the in vitro results, western blot analysis showed a decrease of extracellular exosomes (CD63 and CD9) both in the brain and kidney of KIBRA-KO mice compared with their WT counterparts (Fig. 3b, d). Accordingly, NTA of the exosome pellets isolated from the same volume of serum showed that the number of exosomes isolated from KIBRA-KO mice was only about 52% of that from the WT mice, whereas the mean size of exosomes was not affected (Fig. 3e, f). These results suggested that exosome secretion in peripheral blood serum was impaired in KIBRA-KO mice.

To further visualize the detailed ultrastructure of MVBs in the hippocampus of KIBRA-WT and -KO mice, electron microscopy was used to quantify the number of ILVs and MVBs. The results showed a ~60% increase in the number of MVBs per cell in KIBRA-KO mice compared with their WT littermates (Fig. 3g, h). Meanwhile, the number of ILVs per cell profile and the number of ILVs per MVB increased by ~120% and ~40%, respectively (Fig. 3g–j). Overall, these data strongly support the role of KIBRA in the regulation of exosome secretion as well as the phenotype of MVBs in vivo.

**Depleting KIBRA promotes degradation of Rab27a.** To investigate the molecular mechanisms by which KIBRA regulates exosome secretion, we performed mass spectral (MS) analysis and an isobaric tag for relative and absolute quantitation (iTRAQ) assay to screen the differentially expressed proteins in the brains of KIBRA-KO mice compared with their WT littermates. The mechanisms of exosome biogenesis and secretion have been extensively studied[3,28,29]; thus we analyzed those proteins reported to be involved in exosome formation or secretion. As shown, Rab27a was the most significantly changed protein with the maximum fold change (FC) among all proteins that were statistically significant (P < 0.05) (Fig. 4a and Supplementary Table 1).

To further verify the protein profiles, we compared the levels of Rab27a in the cortex and hippocampus of KIBRA-KO and -WT

mice by immunofluorescence analysis. The results showed that depleting KIBRA resulted in a dramatic decrease of Rab27a expression in the cortex and hippocampus area (Fig. 4b). In addition, western blot analysis indicated that Rab27a decreased significantly in the cortex, hippocampus, kidney, muscle, and liver of KIBRA-KO mice compared with those in their WT counterparts (Fig. 4c, d). Interestingly, the tissue specificity of Rab27a was extremely similar with that of KIBRA, which is predominantly expressed in the brain and kidney (Fig. 4c, d and Supplementary Fig. 1). The real-time PCR results showed that the Rab27a mRNA level remained unchanged in the kidney, muscle, and liver of KIBRA-KO mice or even increased in the cortex and hippocampus (Fig. 4e), which was probably due to a compensatory mechanism for excessive degradation of Rab27a protein. Thus it is reasonable to believe that the decrease in the Rab27a protein level in KIBRA-KO mice is due to increased degradation rather than decreased synthesis of Rab27a.

**KIBRA prevents Rab27a from being ubiquitinated.** To explore the Rab27a degradation mechanisms, KIBRA-KD and Ctrl-KD cells were treated with the protein synthesis inhibitor (cycloheximide (CHX)), the proteasome inhibitor (Lac), or the lysosome inhibitor (Baf). Western blot analysis showed that Rab27a was enormously degraded when treated with CHX for 12 h, and this degradation was restored by the proteasome inhibitor Lac but not by the lysosome inhibitor Baf (Fig. 5a, b). Furthermore, IP experiments showed that Rab27a was more easily ubiquitinated when KIBRA was depleted, and the Lac treatment dramatically increased the levels of ubiquitinated Rab27a (Fig. 5c, d). These results indicate that Rab27a was degraded mainly through the ubiquitin–proteasome pathway.

To further confirm the direct interaction between KIBRA and Rab27a, we performed a cross-IP experiment. 293T cells were cotransfected with GFP-KIBRA and DsRed-Rab27a vectors; 48 h after transfection, the cells were lysed and subjected to pre-clearance with control IgG and then incubated with anti-KIBRA or anti-Rab27a antibodies for IP. The IP products were blotted with anti-Rab27a or anti-KIBRA antibodies in a crossed manner. As shown in Fig. 5e, immunoprecipitating KIBRA with anti-KIBRA antibody pulled down not only KIBRA but also Rab27a. As a control, neither protein was detected in the IgG–immunoprecipitated complex. Similarly, immunoprecipitating Rab27a with an anti-Rab27a antibody pulled down not only Rab27a but also KIBRA (Fig. 5f). Furthermore, immunofluorescent staining of Rab27a in 293T cells transiently transfected with GFP-tagged KIBRA showed considerable co-localization of Rab27a (red) and KIBRA (green) (Fig. 5g). The cross-IP and immunofluorescent results collectively suggest that Rab27a and KIBRA interact directly with each other.

Taken together, these data indicate that Rab27a is degraded mainly through the ubiquitin–proteasome pathway rather than the lysosome pathway. Rab27a becomes stabilized through an interaction with KIBRA and therefore is free from being ubiquitinated. In contrast, depleting KIBRA leads to increased

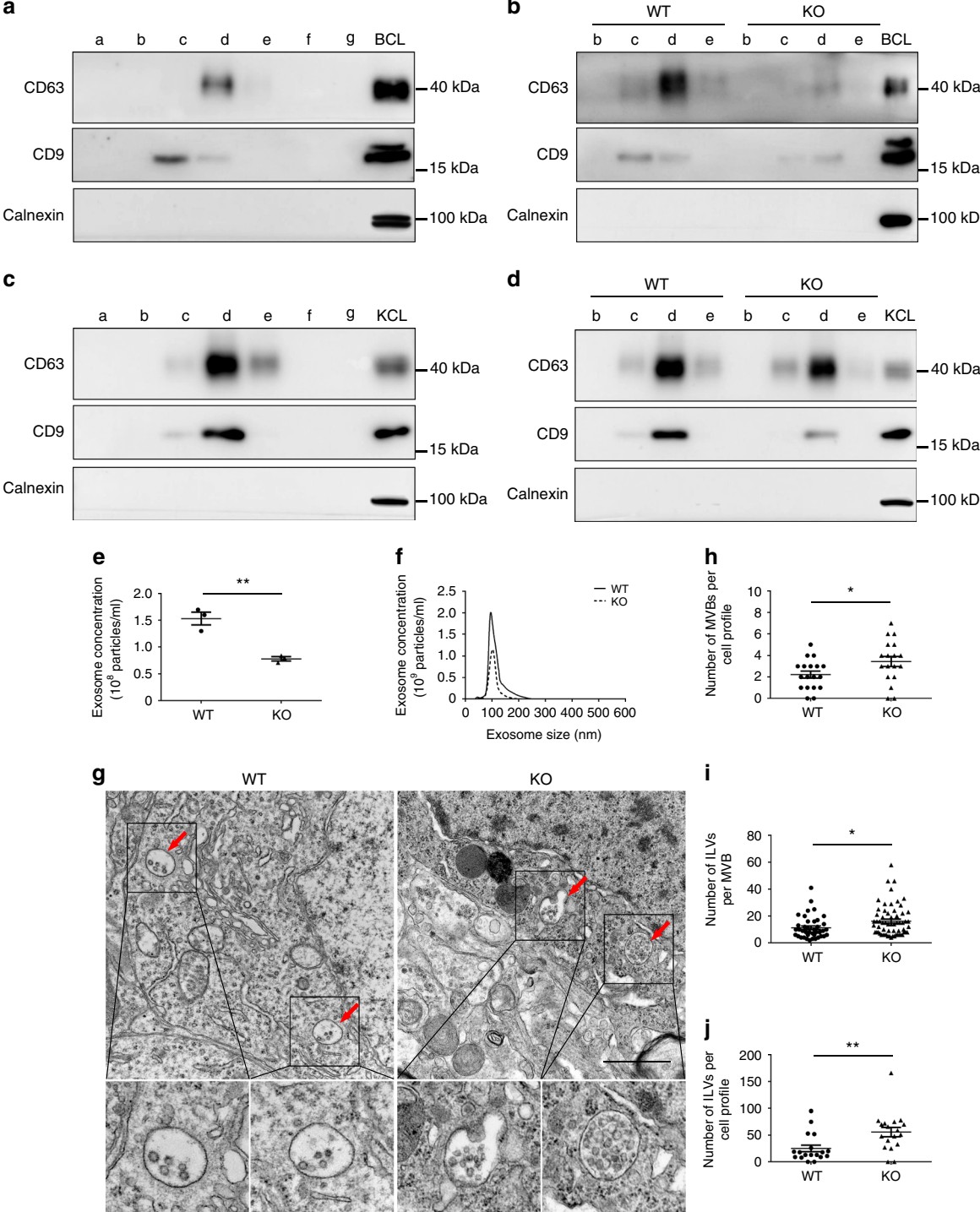

**Fig. 3** KIBRA regulates exosome secretion and the multivesicular body (MVB) phenotypes in vivo. **a, c** Sucrose step gradient fractions a–g isolated from the extracellular space of the brain (**a**) or kidney (**c**) from wild-type (WT) mice were blotted for the exosomal markers CD63, CD9, and Calnexin. The brain cell lysates (BCL) and kidney cell lysates (KCL) were used as controls, respectively. **b, d** Fractions b–e isolated from the extracellular space of the brain (**b**) or kidney (**d**) from WT and KIBRA-KO mice were blotted for CD63, CD9, and Calnexin. **e** Quantification of nanoparticle tracking analysis (NTA) experiments of exosomes isolated from the same volume of serum from KIBRA-KO and WT mice. **f** Representative NTA traces of exosomes isolated from the same volume of serum from KIBRA-KO and WT mice. **g** Representative electron microscopic images of hippocampus area in KIBRA-KO and WT mice. Scale bar = 1 μm. Zooms (lower panels) were ×4.0. Red arrows indicate MVBs containing typical intraluminal vesicles (ILVs). **h** The number of MVBs per cell profile. **i** The number of ILVs per MVB. **j** The number of ILVs per cell profile. ILVs and MVBs in 18 profiles of different cells were counted in a blind manner and only MVBs containing typical ILVs were counted. The quantification results were plotted as dot plots, showing the mean ± SE. *$P < 0.05$, **$P < 0.01$ as determined by two-tailed $t$ test

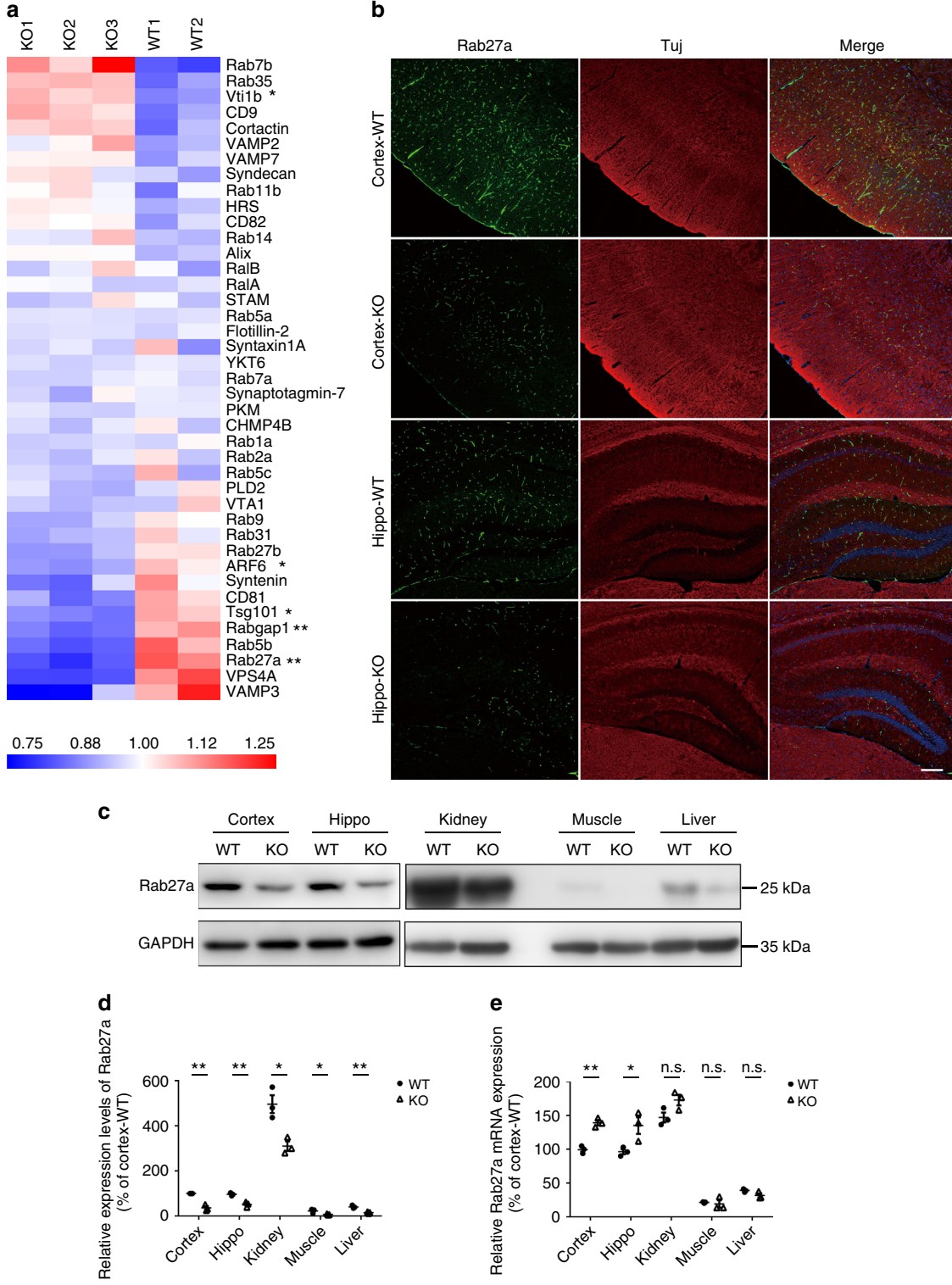

**Fig. 4** Depleting KIBRA promotes the degradation of Rab27a. **a** Heat map of proteins involved in exosome biogenesis or secretion by mass spectrometry in the cortex and hippocampus of three KIBRA-KO mice and two wild-type (WT) controls. Proteins are arranged according to fold change values. Upregulated and downregulated proteins are indicated by red and blue hues, respectively. Color intensity indicates the expression levels of proteins as displayed. **b** Rab27a was detected by immunofluorescent staining in the cortex and hippocampus of KIBRA-KO mice and WT controls ($n = 3$ in each group). Scale bar = 200 μm. **c** Western blot was performed to determine the Rab27a protein levels in the cortex, hippocampus, kidney, muscle, and liver of KIBRA-KO mice and WT controls ($n = 3$ in each group). **d** Quantification of Rab27a levels in different tissues of KIBRA-KO and WT mice in three independent experiments. **e** Real-time PCR was performed to determine the mRNA levels in different tissues of KIBRA-KO and WT mice ($n = 3$ in each group). The quantification results were plotted as dot plots, showing the mean ± SE. *$P < 0.05$, **$P < 0.01$. n.s., not significant ($P > 0.05$) as determined by two-tailed $t$ test

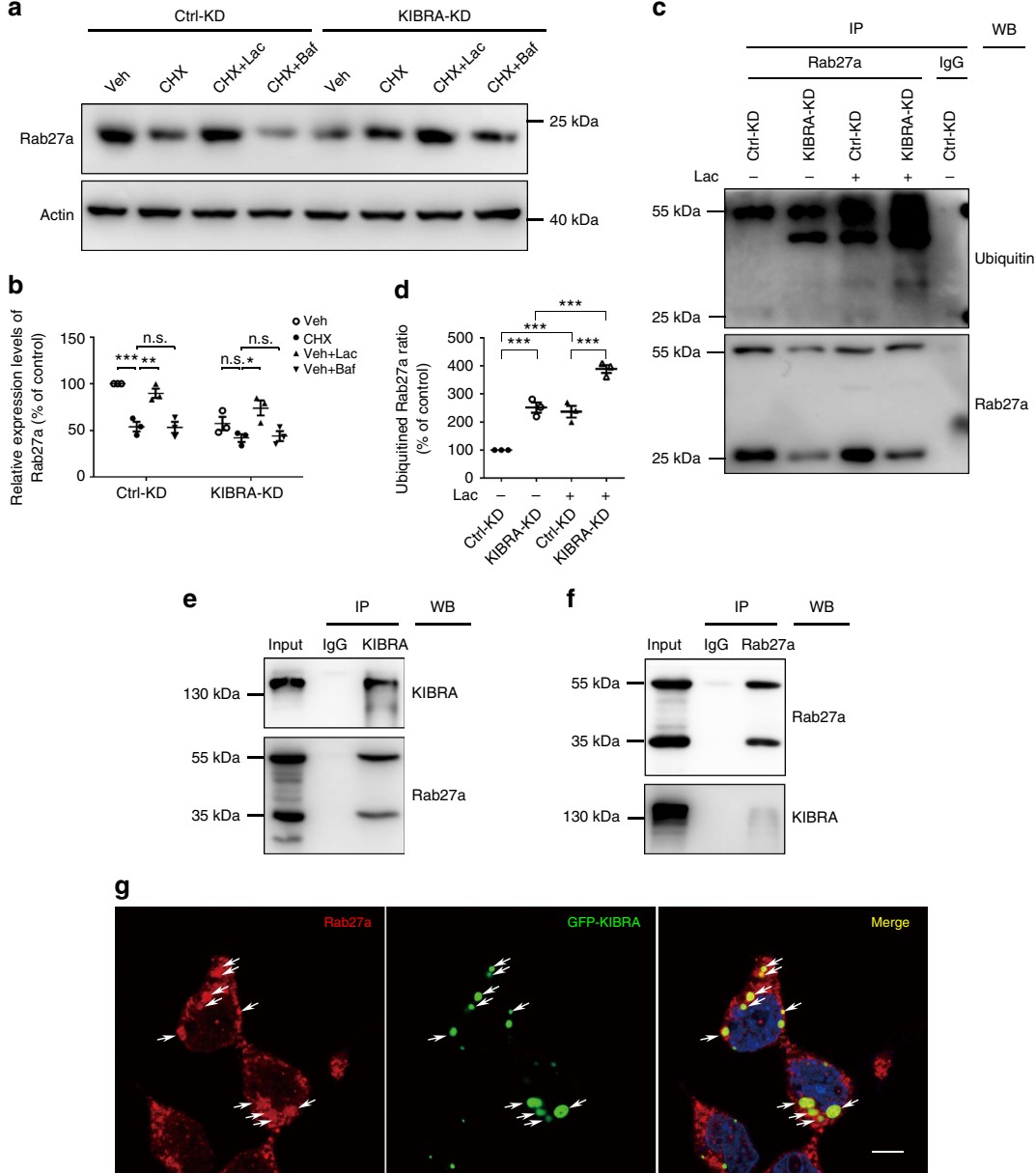

**Fig. 5** KIBRA stabilizes Rab27a and prevents it from being ubiquitinated. **a** Western blot was performed to determine the level of Rab27a in KIBRA-KD and Ctrl-KD cells treated with cycloheximide (CHX) for 12 h to inhibit protein synthesis. Meanwhile, the medium was supplemented with the proteasome inhibitor lactacystine (Lac) or the lysosome inhibitor bafilomycin A1 (Baf). Cells treated with vehicle were used as a control. **b** Quantification of Rab27a levels in three independent experiments. **c** KIBRA-KD and Ctrl-KD cells were treated with Lac or vehicle for 12 h before being lysed and immunoprecipitated using anti-Rab27a antibody or IgG. Ubiquitinated Rab27a levels were determined by western blotting with the anti-ubiquitin antibody. **d** Quantification of ubiquitinated Rab27a levels in three independent experiments. **e** 293T cells were cotransfected with GFP-KIBRA and DsRed-Rab27a. The cells were lysed 48 h after transfection and immunoprecipitated using anti-KIBRA antibody or IgG. The association between KIBRA and Rab27a was determined by western blotting with the anti-Rab27a antibody. **f** 293T cells were cotransfected with GFP-KIBRA and DsRed-Rab27a. Cells were lysed 48 h after transfection and immunoprecipitated using anti-Rab27a antibody or IgG. The association between KIBRA and Rab27a was determined by western blotting with the anti-KIBRA antibody. **g** 293T cells were transiently transfected with GFP-tagged KIBRA (green). After 24 h, the cells were fixed and stained with antibody against Rab27a (red). Scale bar = 10 μm. White arrows indicate the small punctate structures where KIBRA and Rab27a were co-localized. The quantification results were plotted as dot plots, showing the mean ± SE. *P < 0.05, **P < 0.01, ***P < 0.001. n.s., not significant (P > 0.05) as determined by the one-way analysis of variance test

proteasomal degradation of Rab27a, which, in turn, suppresses exosome secretion.

**Overexpression of Rab27a rescues impaired exosome secretion**. To further explore whether KIBRA regulates exosome secretion via Rab27a, KIBRA-KD and Ctrl-KD cells were transfected with DsRed-Rab27a plasmids, and G418 was used to select the

neomycin-resistant transformants. Overexpression of Rab27a and KD of KIBRA were confirmed by western blot analysis (Fig. 6a–c). The cells were cultured in DMEM medium containing 10% exosome-depleted FBS for 48 h, and the exosomes were isolated for further detection.

Consistent with the above results, the total amount of exosomal proteins decreased significantly in KIBRA-KD cells compared

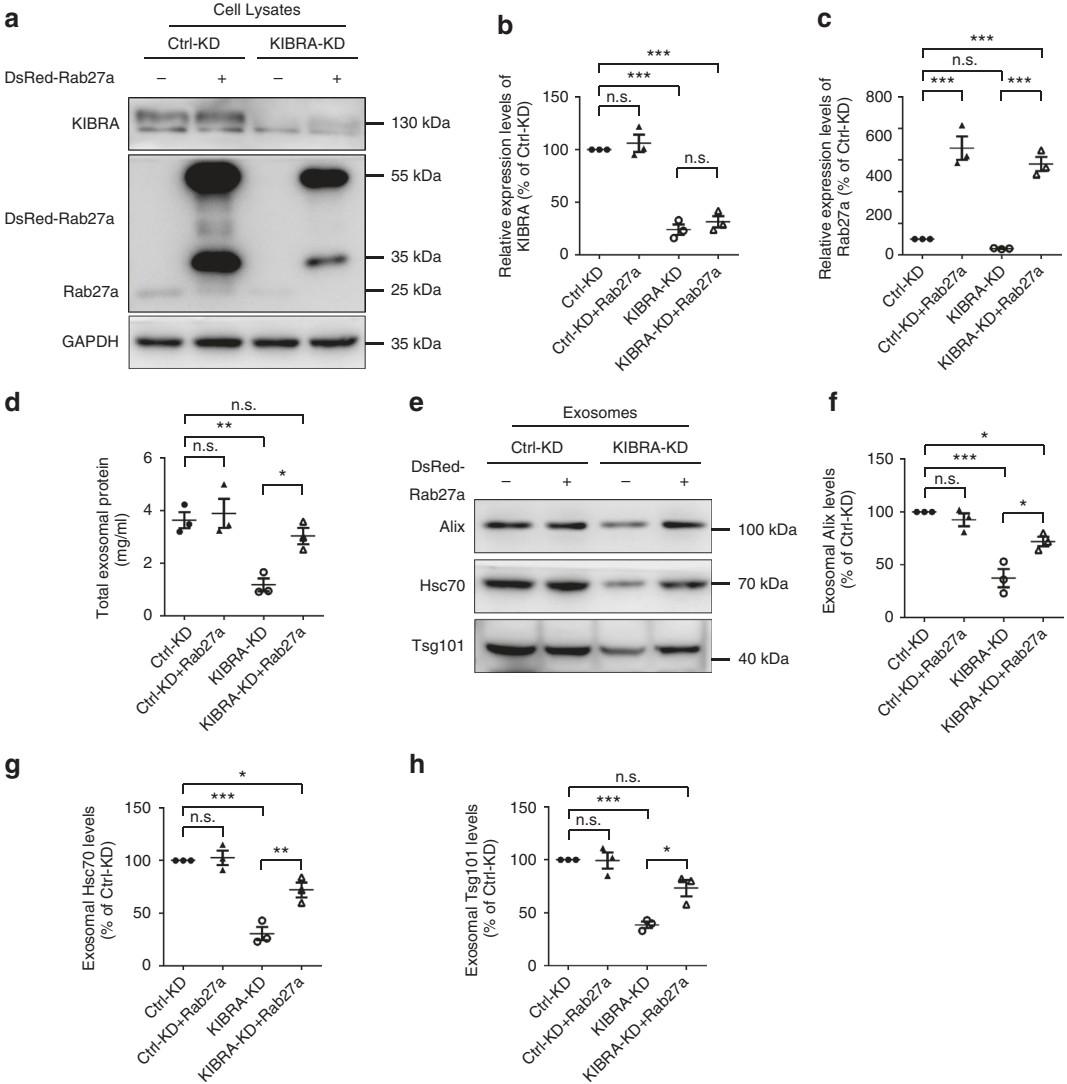

**Fig. 6** Overexpression of Rab27a rescues the impaired exosome secretion induced by depleting KIBRA. **a** Western blot was performed to determine the protein levels of KIBRA and Rab27a in KIBRA-KD and Ctrl-KD cells transfected with plasmids encoding Rab27a or not. **b**, **c** Quantification of KIBRA (**b**) and Rab27a (**c**) levels in three independent experiments. **d** Concentration of exosomal proteins in KIBRA-KD and Ctrl-KD cells transfected with DsRed-Rab27a or not. **e** Western blot analysis of extracellular vesicles purified from equal numbers of KIBRA-KD and Ctrl-KD cells transfected with plasmids encoding Rab27a or not. **f–h** Quantification of Alix (**f**), Hsc70 (**g**), and Tsg101 (**h**) levels in three independent experiments. The quantification results were plotted as dot plots, showing the mean ± SE. *$P < 0.05$, **$P < 0.01$, ***$P < 0.001$. n.s., not significant ($P > 0.05$) as determined by the one-way analysis of variance test

with Ctrl-KD cells. However, the exosomal protein level increased significantly when Rab27a was overexpressed in KIBRA-KD cells (Fig. 6d). Immunoblotting of exosomal markers (Alix, Hsc70, and Tsg101) showed similar results (Fig. 6e–h). These data indicate that overexpressing Rab27a rescues impaired exosome secretion in KIBRA-KD cells. However, exosome secretion did not increase in Ctrl-KD cells, even though Rab27a was overexpressed. Generally, there are two potential explanations. One is that Rab27a is sufficient for MVBs to transport or dock at PM in Ctrl-KD cells, and redundant exogenous Rab27a is useless for exosome secretion. Another explanation is that, although Rab27a levels increased significantly when Rab27a-expressing plasmids were transfected into KIBRA-KD cells, the levels of Rab27a effector proteins remained unchanged. As Rab GTPases control membrane trafficking by recruiting their specific effectors onto membrane surfaces to improve organelle motility or vesicles docking[30], the limited Rab27a effector proteins possibly inhibited the increase in exosome secretion.

## Discussion

The function of EVs to remove excess molecules or mediate cell-to-cell communication is an extremely complex process that requires precise regulation of its secretion. Despite considerable progress identifying several critical controllers of exosome biogenesis and secretion, little is known about the platform or the partner of exosome regulators. In this study, we provide evidence that depleting KIBRA dramatically decreased exosome secretion both in vitro and in vivo, which was accompanied by down-regulation of the Rab27a protein rather than mRNA, a small GTPase reported to regulate MVB docking to the PM. The elevated degradation of Rab27a in the absence of KIBRA was due to its high level of ubiquitination. In addition, KIBRA and Rab27a showed an extremely similar tissue specificity that was predominantly expressed in the brain and kidney, but not in the liver or muscle. The immunofluorescence staining and cross-IP experiment further revealed the co-localization and interaction of KIBRA with Rab27a. In summary, KIBRA functions as an

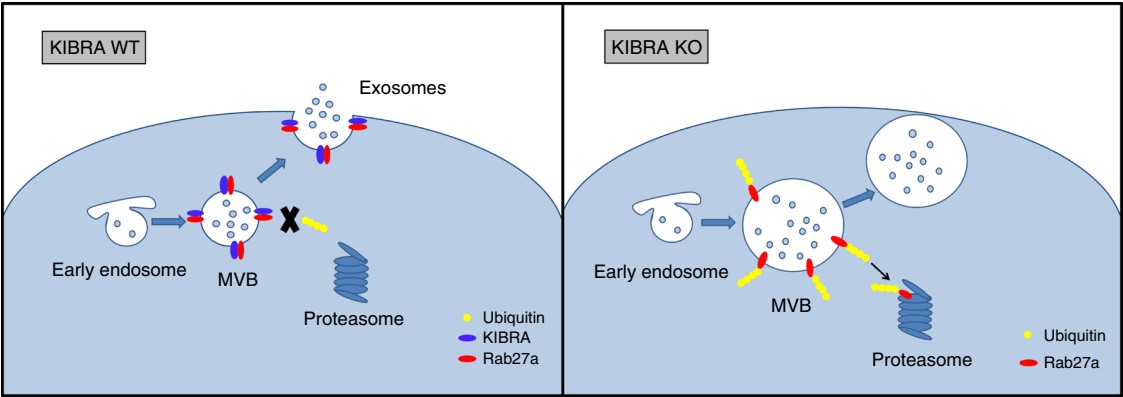

**Fig. 7** A schematic representation of a model for the role of KIBRA in the regulation of exosome secretion. KIBRA stabilizes Rab27a and keeps it from being ubiquitinated. Depleting KIBRA increased Rab27a proteasomal degradation, thus inhibiting exosome secretion

adaptor-like protein and stabilizer for Rab27a, which, in turn, controls small EV secretion in vitro and in vivo (Fig. 7).

EVs are nanoscale size bubble-like membranous structures released by most cell types. There are different types of EVs: microvesicles, which are released by evagination of the PM; apoptosomes, which are released from cells undergoing apoptosis; and exosomes, which are generated by the fusion of MVBs with the PM and the release of ILVs into the extracellular fluid. In our study, we isolated small EVs by sequential centrifugation, i.e., three centrifugations at lower speeds, followed by ultra-centrifugation at $100,000 \times g$ for 70 min. Although this pellet was enriched with exosomes, it probably also contained different vesicle populations of similar sizes. Thus we used the term "small EVs" to indicate isolated vesicles of unknown origin within the cells, and exosomes for vesicles generated through the fusion of MVBs with the PM. In addition, we isolated and purified EVs from the extracellular space of the brain and kidney tissues by sucrose density gradient centrifugation that proved to be endosome-derived exosomes and the integrity of the vesicles could be maintained[31].

KIBRA is a well-known upstream regulator of the conserved Hippo-YAP signaling pathway, which is implicated in cell growth and apoptosis[32,33]. KIBRA has been reported to function as a growth-suppressive regulator in breast cancer[34] and acute lymphocytic B cell leukemia[35]. In contrast, many reports have validated the positive regulatory role of KIBRA in cell proliferation[36,37]. This duality of suppression or promotion of cell proliferation by KIBRA may be tissue dependent, which requires further investigation. In the present study, KIBRA-KD and -OE cells, as well as their control cells, were seeded and cultured in DMEM medium containing 10% exosome-depleted FBS for 48 h prior to exosome isolation. Interestingly, the numbers of cells were similar among these groups 24 and 48 h after seeding, ruling out the possibility that KIBRA regulates exosome secretion in HT22 cells by affecting cell numbers (Supplementary Fig. 5A, B). These discrepant findings might be due to cell-specific regulation or the fact that the follow-up time was too short to detect significant differences.

The Rab family is the largest family of membrane trafficking proteins in mammals, and >60 Rab isoforms are present in humans. Rab GTPases generally function as a molecular switch by cycling between two nucleotide-bound states, such as a GTP-bound active state and a GDP-bound inactive state. Active Rab promotes various steps in membrane trafficking, including vesicle transport and docking as well as fusion of the transport vesicles with acceptor membranes[38]. Using a short hairpin RNA screening approach that targets 59 Rab GTPases in HeLa cells,

Ostrowski et al. identified five possible Rab GTPases (Rab2b, Rab5a, Rab9a, Rab27a, and Rab27b) that were involved in exosome secretion[10]. Rab27a and Rab27b share a common function in MVB docking to the PM but the two Rab27 isoforms have different roles in the exosomal pathway. Rab27a may be required for MVB docking and fusion with the PM, whereas Rab27b may link MVBs to an outwardly directed motor protein[10,39]. Three types of mammalian Rab27a effectors have been described: synaptotagmin-like protein (Slp), Slp homolog lacking C2 domains (Slac2), and Munc13–4[40]. Knockdown of Rab27a effectors leads to a phenotype similar to that of silencing Rab27a[10,41]. In our study, tissue-specific expression of KIBRA was accompanied by tissue-specific expression of Rab27a. However, exosomes are released by virtually all animal tissues. There are two possibilities: either the compensatory effect of Rab27b or the compensatory effect of KIBRA paralogs. First, Rab27b, an isoform of Rab27a, has the ability to bind to most of the Rab27a effectors[42]. Rab27a and Rab27b were originally thought to function redundantly in regulating the secretion pathway by sharing common Rab27 effectors. However, recent evidence indicates that Rab27a and Rab27b may play different roles during exosome secretion[40]. Second, there are two paralogs of KIBRA in humans: WWC2, which has a 49% total sequence identity with KIBRA, and WWC3, which has a 39% total sequence identity with KIBRA but lacks the WW domains. Not all species have all three KIBRA paralogs: no ortholog of KIBRA has been identified in *Caenorhabditis elegans*, only one ortholog of KIBRA has been identified in Drosophila, only KIBRA and WWC3 are present in fish, and only KIBRA and WWC2 occur in mice[43,44]. So far, no report has shown whether WWC2 or WWC3 can also regulate membrane trafficking events.

Evidence has emerged recently that exosomes may play a potential role in neurodegenerative disorders, in which a common central feature is the aggregation and deposition of specific misfolded proteins, such as amyloid-β, tau, prions, and α-synuclein. It has been reported that exosomes containing aggregation prone proteins may facilitate the spread of the misfolded proteins from one to another brain region and thus may accelerate the disease progression[45]. On the other hand, exosomes may help transport excessive or obsolete cellular proteins from the central nervous system to the peripheral organs for degradation[46]. The interaction of KIBRA with Rab27a in controlling exosome secretion may be involved in the initiation and progression of neurodegenerative diseases, although whether KIBRA confers a protective or destructive effect in neurodegenerative diseases remains unclear. A genome-wide screen for memory-related gene variants identified KIBRA to be associated with better memory performance in a

large-scale cognitively normal cohort[47]. However, the relationship of T allele of KIBRA rs17070145 with late-onset Alzheimer's disease remains contradictory[48,49]. In summary, our study suggests that KIBRA controlling the exosome secretion may elucidate other potential function of KIBRA in the brain in the future. Given that EVs are known to play a pivotal role in various cell functions, the exact molecular mechanisms implicated in EV secretion warrant further exploration.

## Methods

**Antibodies and reagents**. The following primary antibodies were used: rabbit polyclonal anti-KIBRA (sc-133374; Santa Cruz Biotechnology, Santa Cruz, CA, USA); mouse monoclonal anti-KIBRA (clone 2A5, provided by Jixin Dong's lab); mouse monoclonal anti-Alix (#2171; Cell Signaling Technology, Danvers, MA, USA); mouse monoclonal anti-Tsg101 (ab83; Abcam, Cambridge, MA, USA); mouse monoclonal anti-CD63 (ab217345; Abcam); rabbit polyclonal anti-calnexin (ab22595; Abcam); mouse monoclonal anti-LBPA (MABT837; Sigma St. Louis, MO, USA); rabbit polyclonal anti-EEA1 (ab2900; Abcam); rabbit polyclonal anti-LC3B (#2775; Cell Signaling Technology); rat monoclonal anti-LAMP2 (ab13524; Abcam); rabbit monoclonal anti-Rab27a (#69295; Cell Signaling Technology); mouse monoclonal anti-beta III Tubulin (ab78078; Abcam); and rabbit monoclonal anti-Hsc70 (ab51052; Abcam). CHX (#2112) was purchased from Cell Signaling Technology; Baf (A8510) was purchased from Solarbio Science & Technology Co., Ltd. (Beijing, China), and Lac (ab141411) was purchased from Abcam. When indicated, the medium contained 40 μg/ml CHX or 20 nM Baf or 10 μM Lac.

**Cell cultures**. HT22 cell line were purchased from Jennio (Guangzhou, China) and the 293T cell line were obtained from Shanghai GeneChem (Shanghai, China). HT22 cell line and 293T cell line were both cultured in DMEM medium (Gibco, Grand Island, NY, USA) supplemented with 10% FBS (Gibco) and 1% penicillin–streptomycin (Beyotime Biotechnology, Beijing, China) in a humidified incubator at 37 °C with 5% CO2. MPC5 cell line, a gift from Dr. Qinglian Wang (Shandong Provincial Hospital affiliated to Shandong University, Jinan, China), was cultured in RPMI-1640 medium (Gibco) supplemented with 10% FBS (Gibco) and 1% penicillin–streptomycin (Beyotime Biotechnology) in a humidified incubator at 37 °C with 5% CO2. All cell lines were routinely tested for mycoplasma contamination.

**CRISPR-Cas9 and sequencing**. To generate KIBRA-KD cells using CRISPR-Cas9 gene editing system, Cas9 vectors and sgRNA targeting the KIBRA gene were cloned into GV370-CMV-hSpCas9-SV40-Puro and GV371-U6-KIBRA sgRNA-SV40-EGFP, respectively. The LV-Cas9 and LV-sgRNA lentiviruses were produced by GeneChem (Shanghai, China). We constructed three sgRNA oligonucleotides and tested the effects of the three sgRNAs on gene silencing prior to the experiments. As shown in Supplementary Fig. 2A, D, all sgRNA oligonucleotides against KIBRA had a similar inhibitory effect and the most efficient was chosen for the following experiments. The sgRNA sequence was as follows: CATCAGTGATGAG TTACCGC. LV-Cas9 was seeded in HT22 cells with a multiplicity of infection (MOI) of 50 after 5–7 days of puromycin (P8032, Solarbio) selection at a final concentration of 3 μg/ml; LV-sgRNA was seeded, and the empty vector was used as controls. After 5–7 days, the KIBRA-KD cells and their control cells were harvested, which are referred to as KIBRA-KD and Ctrl-KD cells, respectively.

To verify the effect of KIBRA in MPC5 cells, LV-Cas9 was seeded in MPC5 cells with an MOI of 20 after 5–7 days of puromycin selection at a final concentration of 0.6 μg/ml; LV-sgRNA was then seeded, and the empty vector was used as a control. Another 5–7 days later, the KIBRA-KD cells and their control cells were harvested, which are referred to as MPC5-KD and MPC5-Ctrl cells, respectively.

**Lentivirus-mediated overexpression of KIBRA**. To construct lentiviral vectors expressing FLAG-tagged KIBRA, murine full-length KIBRA cDNA was used as a PCR template to clone KIBRA into GV341-Ubi-KIBRA-3FLAG-SV40-Puro. The primers used for amplification were 5′-CCAACTTTGTGCCAACCGGTCGCCAC CATGCCCCGGCCGGAGTTGCCCCTGCCG-3′ (forward) and 5′-AATGCCAAC TCTGAGCTTGACGTCATCTGCAGAGAGAGCTGGGATG-3′ (reverse). The lentiviruses were produced by GeneChem. LV-KIBRA was seeded in HT22 cells with an MOI of 20, and the empty vector was used as controls. After 5–7 days of puromycin selection at a final concentration of 4 μg/ml, the KIBRA-overexpressing cells (referred to as KIBRA-OE) and control cells (referred to as Ctrl-OE) were harvested. The KIBRA-OE cells accounted for >90% of the infected HT22 cells.

**Plasmid construction and transfections**. To produce GFP-tagged KIBRA and the DsRed-tagged Rab27a chimera, murine full-length KIBRA and Rab27a were amplified and cloned into the pEGFP-N1 and pDsRed2-N1 plasmids, respectively. 293T and HT22 cells were grown on glass coverslips and transfected at 50% confluency with GFP-KIBRA or the DsRed-Rab27a plasmid using Lipo2000 transfection reagent (11668030; Thermo Scientific, Waltham, MA, USA) at a ratio of 0.2 μg vector DNA per 1 μl of cationic lipid according to the manufacturer's

instructions. After 48 h of expression, the transfection efficiency in 293T cells was >80%, and 293T cells were harvested for further detection. However, the transfection efficiency of the DsRed-Rab27a plasmid in HT22 cells was only 20–30% after 48 h of expression. Then 800 μg/ml G418 (G8160, Solarbio) was used to select the neomycin-resistant transformants. After G418 selection, transfection efficiency was >70%, and HT22 cells were harvested for further detection.

**Animals**. Wwc1tm1.1Rlh mice were purchased from the Jackson Laboratory (No. 024415; Bar Harbor, ME, USA). KIBRA was completely knocked out in all tissues of this strain. The mice were housed in individually ventilated cages on a 12 h light–dark cycle at 21–23 °C and 40–60% humidity. Mice were allowed free access to an irradiated diet and sterilized water. All experiments were performed on homozygous mice (KIBRA-KO) and their age- and gender-matched wild-type littermates (KIBRA-WT) derived from heterozygous breeding. n values in each figure legend reflect the number of animals used for the statistical analysis. All experimental procedures were performed in accordance with the protocols approved by the Institutional Animal Care and Research Advisory Committee of Shandong University, Jinan, Shandong. All efforts were made to reduce the number of animals used and to minimize animal suffering.

**Western blot**. Brain tissues, cells, and exosome fractions were lysed in Mammalian Protein Extraction Reagent (78501, Thermo Scientific) containing a protease inhibitor cocktail (04693132001, Roche Diagnostic, Indianapolis, IN, USA). The proteins were separated by 8–12% acrylamide/bisacrylamide gel electrophoresis and transferred to PVDF membranes (Millipore, Billerica, MA, USA). The proteins were probed with primary antibodies followed by an horseradish peroxidase (HRP)-conjugated secondary antibody. Immunodetection was carried out with the Immobilon Western Chemiluminescent HRP substrate (Millipore). Normalization was conducted by blotting the same samples with an antibody against Actin or GAPDH. All uncropped data are shown in Supplementary Fig. 6.

**Co-immunoprecipitation (co-IP)**. Co-IP was performed using the Pierce Co-Immunoprecipitation Kit (26149, Thermo Scientific) according to the manufacturer's instructions. Briefly, cells were lysed with lysis buffer (25 mM Tris, 150 mM NaCl, 1 mM EDTA, 1% NP-40, and 5% glycerinum, pH 7.4) for 10 min on ice. The lysates were cleared by centrifugation at 13,500 × g for 10 min at 4 °C and then immunoprecipitated individually with anti-Rab27a antibody, anti-KIBRA antibody, or normal IgG. After the elution, the proteins were lysed in RIPA lysis buffer for the western blot analysis.

**Exosome isolation**. To isolate exosomes from the cellular supernatant, 80% confluent cells were rinsed with phosphate-buffered saline (PBS) and refreshed with DMEM containing 10% exosome-depleted FBS (bovine exosomes were removed by overnight centrifugation at 100,000 × g). After a 48-h incubation, an equivalent volume of culture medium conditioned by an equivalent number of cells was collected, and the exosomes were isolated at 4 °C by sequential centrifugation[50]. Briefly, the medium was centrifuged at 300 × g for 10 min to remove cells. The supernatant was centrifuged at 2000 × g for 10 min and then at 10,000 × g for 30 min. The pellets were collected and referred to as the 2K pellet and the 10K pellet, respectively. The resulting supernatant was filtered through a 0.22-μm filter, and the exosomes were pelleted by ultracentrifugation at 100,000 × g (Beckman Type 90 Ti) for 70 min. The exosome pellet was washed in cold PBS and collected by ultracentrifugation again at 100,000 × g (Beckman Type 90 Ti) for 70 min. Finally, the exosome pellet was resuspended in PBS or lysis buffer before further analysis.

To analyze the effect of KIBRA on exosome secretion in vivo, we isolated exosomes from the brain and kidney extracellular space of 5-month-old KIBRA-KO mice and their age- and gender-matched WT controls[31]. Fresh or previously frozen murine hemi-brains or the kidney were dissected and treated with 1.5 mg/ml collagenase D (11088858001, Roche) in Hibernate A solution (A1247501, Thermo Scientific). The tissues were incubated in a water bath at 37 °C for 20 min. After incubation, the tissues were returned to ice immediately, and the protease inhibitor cocktail (04693132001, Roche) was added to a final concentration of 1×. The dissociated tissue was sequentially centrifuged following the same sequential centrifugation steps as those used for the cellular supernatant to collect the exosome pellet. Brain and kidney cells recovered from the first 300 × g pellet were lysed to perform a western blot as controls for the exosomes. The washed exosome pellet was resuspended in 2 ml of 0.95 M sucrose solution and inserted inside a sucrose step gradient column (six 2-ml steps starting from 2.0 M sucrose up to 0.25 M sucrose in 0.35 M increments)[51]. The sucrose step gradient was centrifuged at 200,000 × g (Beckman SW 41 Ti) for 16 h at 4 °C. A total of seven fractions (a–g) were collected, diluted in cold PBS, and centrifuged at 100,000 × g at 4 °C for 70 min. Seven sucrose gradient fraction pellets were resuspended in 30 μl lysis buffer before further western blot analysis.

**Transmission electron microscopy**. A 10-μl aliquot of freshly isolated exosomes were allowed to dry on a formvar/carbon-coated copper grid for 10 min and were fixed in 3% glutaraldehyde for 10 min, rinsed in water, and contrasted in a uranyl acetate (4%)/methylcellulose (1%) mix for 10 min at room temperature to

negatively stain the exosomal fractions. Then the samples were observed immediately at 80 kV with a JEOL-1200EX electron microscope (JEOL, Tokyo, Japan).

KIBRA-KD and Ctrl-KD cells were rinsed in 0.1 M PBS and fixed with 2.5% glutaraldehyde overnight at 4 ℃ for the conventional electron microscopic analysis. Anesthetized mice were transcardially perfused with 50 ml of 2% glutaraldehyde and 2.5% paraformaldehyde in 0.1 M PBS at 4 ℃ to collect the hippocampal tissues. Then the hippocampal tissues were excised immediately and cut into 1 mm³ pieces in cold fixative. Both the cells and tissue specimens were post-fixed for 1.5 h with 1% osmium tetroxide in phosphate buffer, dehydrated through a graded ethanol series, and embedded in Epon812. Ultrathin sections (70 nm) were prepared, stained with uranyl acetate and lead citrate, and examined by electron microscopy.

**Nanoparticle tracking analysis**. The number and size of the exosomes were directly tracked using the NS300 instrument (Malvern Instruments Ltd., Worcestershire, UK) equipped with a 488 nm laser and a high-sensitivity sCMOS camera. In this analysis, particles are automatically tracked and sized based on Brownian motion and the diffusion coefficient. The exosome pellets were resuspended and diluted in PBS to obtain a concentration within the recommended range ($2 \times 10^8 – 1 \times 10^9$ particles/ml) and vortexed for 1 min. The samples were loaded into the sample chamber at ambient temperature. Three 30-s videos were acquired for each sample. The videos were subsequently analyzed with the NTA2.3 software, which identified and tracked the center of each particle under Brownian motion to measure the average distance the particles moved on a frame-by-frame basis.

**Immunofluorescence**. Cells were grown on glass coverslips and harvested at 80% confluency to prepare the cell slides for immunofluorescent staining. Anesthetized juvenile mice were transcardially perfused with 50 ml of 0.1 M PBS and 50 ml of 4% paraformaldehyde to prepare the frozen brain sections. After dissection, the brains were sequentially soaked in 15 and 30% sucrose/0.1 M PBS for 24 h. After dehydration, the brains were quickly frozen and coronally sectioned at 10 μm thickness.

Slides of cells and frozen sections of brains were fixed with 4% paraformaldehyde, permeabilized with 0.5% Triton X-100 for 10 min, and probed with primary antibodies diluted 1:200 in PBS overnight at 4 ℃. After washing in PBS, the cells were incubated with fluorescent-conjugated secondary antibodies (488 and 594 nm) for 1 h. All samples were treated with DAPI dye for nuclear staining (358 nm). After additional washes in PBS, images were captured using a LSM780 confocal scanning microscope (Zeiss, Jena, Germany). The images were processed and analyzed with the ZEN 2010 software (Zeiss), ImageJ (National Institutes of Health, Bethesda, MD, USA), and Adobe Photoshop (Adobe Photosystems Inc., La Jolla, CA, USA). A total of 30–50 cells were analyzed per condition in a blind manner, and the experiment was performed independently three times.

**Quantitative real-time PCR**. Total RNA was extracted with TRIzol reagent (Invitrogen, Carlsbad, CA, USA), and complementary DNA (cDNA) was synthesized with the ReverTra Ace qPCR RT Kit (FSQ-101; Toyobo, San Jose, CA, USA). qRT-PCR was performed in triplicate with SYBR Green Realtime PCRMaster Mix (QPK-201; Toyobo). The primers for the genes of interest were synthesized by Biosune (Shanghai, China), as follows: 5′-CAAACAGCTTCCAGCTAAGGAC-3′ (forward) and 5′-GAGAACTCTGTGCCTCACCTCA-3′ (reverse) for Rab27a; 5′-CACTCTCTGTGAGCTGAACCT-3′ (forward) and 5′-GCGGACACACAGGCTACTTT-3′ (reverse) for KIBRA; and 5′-GGACACTGAGCAAGAGAGGC-3′ (forward) and 5′-TTATGGGGGTCTGGGATGGA-3′ (reverse) for GAPDH; GAPDH was used to normalize mRNA expression. FCs were calculated by the $2^{-\Delta\Delta CT}$ method.

**Mass spectrometric analysis**. Total proteins were extracted and digested with Trypsin Gold peptides (Promega, Madison, WI, USA) at 37 ℃ for 16 h. After desalting, the peptides were labeled with iTRAQ reagents (iTRAQ® Reagent-8PLEX Multiplex Kit, Sigma), following the manufacturer's instructions (AB Sciex, Foster City, CA, USA). One unit of labeling reagent was used for 0.1 mg of peptides. Differently labeled peptides were mixed equally and then desalted on 100 mg SCX columns (Strata-x-c, 8B-S029-EBJ; Phenomenex, Torrance, CA, USA). The iTRAQ-labeled peptide mix was fractionated using a C18 column (BEH C18 4.6 × 250 mm, 5 μm; Waters, Milford, MA, USA) on a Rigol L3000 high-performance liquid chromatography system operating at 1 ml/min. The column oven temperature was set to 50 ℃. Mobile phases A (2% acetonitrile, 20 mM NH₄FA, adjusted to pH 10.0 using NH₃·H₂O) and B (98% acetonitrile, 20 mM NH₄FA, adjusted to pH 10.0 using NH₃·H₂O) were used to develop a gradient elution. The eluent was dried under a vacuum and reconstituted in 20 μl of 0.1% (v/v) formic acid and 3% (v/v) acetonitrile in water for subsequent analyses. The fractions were dissolved in loading buffer and separated on a C18 column (150 μm inner diameter, 360 μm outer diameter × 15 cm, 1.9 μm C18, Reprosil-AQ Pur; Dr. Maisch, Ammerbuch, Germany). The Q-Exactive HF-X mass spectrometer was operated in positive polarity mode with a capillary temperature of 320 ℃. Full MS scan resolution was set to 60,000 with an AGC target value of 3,000,000 and a scan range of 350-1500 $m/z$.

The resulting spectra from each fraction were searched separately against the Uniprot database using the Proteome Discoverer 2.2 search engine (PD 2.2; Thermo Scientific).

**Statistical analysis**. Results are expressed as mean ± standard error (SE) and analyzed by the GraphPad Prism 6.0 software (GraphPad Software Inc., La Jolla, CA, USA). Technical as well as biological triplicates of each experiment were performed. Comparison between two groups was performed by Student's $t$ test. Multiple group comparisons were determined using one-way analysis of variance. A $P$ value <0.05 was considered statistically significant.

## Data availability
The mass spectrometric proteomics data have been deposited to the ProteomeXchange Consortium via the PRIDE partner repository with the dataset identifier PXD012430. All other data that support the conclusions of this study are available from the corresponding author upon request.

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

## Acknowledgements

This work was funded in part by grants from the National Key R&D Program of China (grant no.: 2017YFC1310100), the National Natural Science Foundation of China (grants nos.: 81772448, 81501099, and 81861138008), and the Key R&D Program of Shandong Province (grant no.: 2016ZDJS07A11).

## Author contributions

Y.D. and Y.W. designed the study. L.S., S.T., X.H., Z.J., L.D., C.L. and X.L. performed the experiments and analyzed data. Y.W., J.D. and Y.D. contributed to the materials for the study. L.S., S.T. and Y.W. wrote the manuscript. L.S., S.T., Y.W., C.Q. and Y.D. contributed to critical discussions and revisions.

## Additional information

**Competing interests:** The authors declare no competing interests.

