## [Peer Review File · Nature Communications]

Reviewers' comments:

Reviewer #1 (Remarks to the Author):

In this manuscript, authors demonstrated that KIBRA regulates exosome secretion both in vitro and in vivo. Knockdown or overexpression of KIBRA in the hippocampal neuronal cells (HT22 cells) leads to decrease or increase of exosome secretion, without altering exosome size and morphology. In KIBRA-knockout mice, KIBRA depletion strongly increases the size and number of MVBs and promotes the fusion of MVBs with lysosomes. The comparison of protein profiles between KIBRA knockout and wild-type mice brain reveals a significant decrease of Rab27a, which is a small GTPase reported to regulate MVBs docking to the plasma membrane. Inhibition of trafficking between MVBs and lysosomes with a specific inhibitor in KIBRA-knockdown cells could restore Rab27a protein level, indicating that KIBRA depletion induced lysosomal degradation of Rab27a through MVB/lysosome pathway. In addition, overexpression of Rab27a in KIBRA-knockdown cells rescues exosome secretion. The overall observation is very interesting. There are two major concerns: 1) whether KIBRA regulates exosome secretion is a general phenomenon or tissue specific phenotype? If the authors would like to claim this is a general phenomenon, more than one cell line or tissues need to be test; 2) the mechanism how loss of function KIBRA induced degradation of Rab27a was not studied.

Specific points:

Authors knockdown or overexpressed KIBRA in HT22 cells, however, the endogenous expression level of KIBRA was dramatically different in supplemental Figure 1A & B. Can authors explain this discrepancy?

Figure 2, a second marker other than CD63 needs to be included to confirm the authors' conclusion.

Figure 3, is it brain specific KIBRA knockout in Wwc1tm1.1Rlh mice? If not, can authors detect exosome secretion difference in the blood compare to wild type mice?

Although authors claimed the GFP-KIBRA punctae displayed a significant, but not complete, colocalization with the MVBs marker CD63. However, Figure 4A showed GFP-KIBRA exclusively localized with MVBs.

Reviewer #2 (Remarks to the Author):

The article by Song et al describes a novel role of the protein KIBRA (gene name WWC1) in regulation of secretion of exosomes, i.e. small Extracellular vesicles (EVs) originating as intraluminal vesicles of multivesicular endosomes. Using complementary models of KIBRA down-regulation (CRISPR/Cas9 in cells and KO mice) and overexpression, the authors further show that absence of KIBRA leads to increased number of MVBs and their ILVs, and decreased level of exosome secretion. The authors observe an increased colocalization of MVB markers (CD63) with lysosome markers (Lamp2) and conclude that MVBs fuse more with lysosomes in the absence of KIBRA. By quantitative proteomics, they observe deregulation of the level of several intracellular proteins previously described as involved in exosome formation or secretion, and follow more precisely the fate of Rab27a : this protein is less abundant in KIBRA KO/KD cells, due to enhanced degradation.

The authors end up with a model of Rab27a degradation induced by fusion of MVBs with lysosomes in the absence of KIBRA.

The work is interesting, relatively novel : roles for KIBRA in exosome or EV secretion, or in RAB27 protein stability have never been described, although other roles for KIBRA in intracellular trafficking and exocytosis have been published (ref 21 Traer et al 2007, and Yoshihama et al. 2011. Curr Biol 21:705, not quoted, slightly contradictory with this work). It convincingly shows that KIBRA regulates stability of several intracellular trafficking protein, including Rab27a, which (probably together with modified expression of other Rabs and ESCRT) lead to decreased exosome secretion. It does not, however, lead to conclusions on more functional aspects of EV and/or exosome secretion.

My first concern is that, due to its pleiotropic effect, It is very likely that KIBRA also regulates

secretion of non MVB-derived small extracellular vesicles, co-isolated in the ultracentrifugation pellets (the protocol used by the authors is now known to co-isolate multiple types of small EVs, and specific markers of MVB-derived small EVs = exosomes are not yet strictly known : see Kowal et al, PNAS 2016, 113 : E968). Therefore, the authors should start their article using the generic term « small EVs », before moving to the term exosomes, when they more specifically analyse MVBs. And they should also analyse the effects of KIBRA KO/KD on secretion of other types of EVs, such as larger EVs recovered by lower speed centrifugation.

My second concern is that the model of Rab27a degradation by lysosomes through fusion of MVB with lysosomes induced in the absence of KIBRA, however, does not satisfy me, because it does not fit with the current knowledge on RAB27A intracellular localisation, and on the relationship between MVB and exosomes. 1) the observed increased MVB/ILV number upon KIBRA KO shown in fig2 is consistent with the previously published effect of Rab27a KO on CD63+ compartments in HeLa cells (Ostrowski et al, ref 10), and the consequent reduced level of exosome secretion, but contradictory with the proposed enhanced fusion of MVB with lysosomes : enhanced fusion of MVB with lysosomes should lead to decreased numbers of MVBs instead of increased numbers. 2) Results of figure 4 (co-localisation of KIBRA, CD63 and Lamp2), cannot be interpreted in terms of fusion of MVB with lysosomes, they only show some increase of Lamp2 and CD63 colocalisation, which could be due to mislocalisation of Lamp2 in MVBs instead : only EM and quantification of intracellular localisation of CD63 and Lamp2 in MVBs vs lysosomes could demonstrate the authors point. 3) In any case, since Rab27a is a cytosolic protein, it is attached to the outer surface of MVB, and will not end up inside lysosomes if the MVB fuse with a lysosome, it will instead remain outside the MVB/lysosome compartment. Most likely, degradation of Rab27a observed by the authors is due to another mechanism of degradation, maybe proteasome-mediated (the authors must evaluate the effect of proteasome inhibitors like lactacystine on Rab27a level in wt and kibra ko cells). BafA1 treatment for 12h is indeed also affecting other intracellular trafficking pathways, than lysosomal acidification (Palokangas, et al. 1998, Mol. Biol. Cell 9, 3561), and its effect on Rab27a may be indirect via other proteins.

Another possible mechanisms would be by direct or indirect association between KIBRA and Rab27a, which would stabilize the latter : the authors should test whether Rab27a is co-precipitated with KIBRA. Alternatively, since this part of the article is not very important for the rest of the message, I would suggest to delete fig4, fig5f-g and fig7.

Technical comments :

1) Figure 1C showing the level of different EV-associated proteins in the isolated ultracentrifuged pellets from KIBRA-KD or -OE cells should also show the levels of these proteins in the cell lysates to determine if the differences in secretion levels are due to impaired secretion of EVs or rather to decreased intracellular levels of the EV markers.

2) EV isolation from brain tissue is performed in a way that separate eliminate fragments of cells, generated during tissue mincing and passing through needles. The authors quote ref 44 for this protocol, but they should be aware of more careful work performed on brain tissue, to isolate EVs and exosomes through gradient-based separation : Vella et al, 2017, J Extracell Vesicles 6, 1348885.

3) The anti-CD63 antibody should show a fuzzy band, due to high level of glycosylation, the sharp bands shown in WB are probably not specific. The WB showing Rab27a bands in figure 6 should show position of both the endogenous Rab27a and the Rab27a-mcherry construct : it is not clear why Rab27a-Cherry overexpression should enhance expression of the endogenous protein (25kDa, fig 6a), nor if the transfected protein or upregulation of the endogenous one is responsible for recovery of the KIBRA-KO phenotype.

4) Suppl Table 1 does not seem right: column 1 = KO vs WT indicates values between 0.8 and 1.2 of FC for all proteins analysed, whereas column 2 = Log2FC gives much more variable values : what is column 1 showing ? the actual values of the 2 or 3 samples and/or the mean \pm sd would be more informative in this column. In Figure 5a, explain the color code. Finally, these analyses should have been done (and would likely be more reproducible) in 3 independent samples of the CTRL-KO and KIBRA-KO cell lines.

5) some typos and grammatical errors could be corrected throughout the manuscript

Reviewer #3 (Remarks to the Author):

Song et al. investigate the role in exosome release played by KIBRA/WWC1, a protein with a poorly understood function that has been implicated to have a scaffolding role in the Hippo signaling pathway. The authors find that knocking down KIBRA expression in cultured cells reduces exosome release and increases the abundance of endosomal multivesicular bodies (MVBs), the latter of which have intraluminal vesicles (ILVs) that correspond to exosomes when MVBs fuse with the plasma membrane. Conversely, KIBRA overexpression is found to increase exosome release, though no measure of MVB abundance under this condition is provided. In-vivo studies of exosomes release in knockout mice lacking KIBRA expression generally support the authors' findings from cultured knockdown cells. By performing mass-spec analysis of wild-type and KIBRA-knockout mice, the authors find that expression of Rab27a is decreased to $\leq 50\%$ normal in the absence of KIBRA expression; further, overexpression of Rab27a in KIBRA-knockdown cultured cells rescues exosome abundance, supporting the authors' proposal that the loss of KIBRA inhibits exosome release through the aberrant degradation of Rab27a, the latter of which appears to occur via lysosomal degradation because Rab27a levels are recovered when cells are treated with the V-ATPase inhibitor, Bafilomycin A1.

In general, this study provides evidence that KIBRA is required for MVB fusion with the plasma membrane, which is necessary for exosome release. It is difficult at this point, however, to claim that KIBRA is regulating this process because the experiments address what happens when KIBRA is absent, which is presumably abnormal for those cells that express KIBRA. Were KIBRA to have a 'switch' function that controls one or more targets involved in MVB-plasma membrane fusion, an active role in controlling exosome release would be more conceivable. Thus, as it stands, the authors' study reveals a requirement for KIBRA, but its role is not evident at this point. For example, does KIBRA directly bind to Rab27? It is puzzling that KIBRA also regulates dynein light chain, which is completely unrelated to exosome trafficking (as far as is known). Can KIBRA be a chaperone for Rab27 and other molecules?

Minor points:

- 1) There are numerous errors in syntax, grammar, and spelling; and the frequency of these errors increases as the manuscript progresses. More careful editing is recommended.
- 2) Results from NTA are presented in figure 1 without a description of this method of analysis or rationale for its use.
- 3) Supplemental figure 1C, D: please clarify if protein or mRNA levels are measured.
- 4) Figure 1C would benefit from showing the abundance of each protein in total cellular extracts from control versus knockdown conditions.
- 5) The increased number of ILVs in MVBs of KIBRA-knockdown cells is intriguing. Do the authors have an idea of why this might be the case?
- 6) Figure 2D: please clarify if '20 cell profiles' refers to profiles of 20 different cells or includes serial sections of the same cells.
- 7) In general, the term 'significant' is liberally used to describe differences in various parameters when comparing cells/mice that have normal versus alter KIBRA expression. More accurate descriptions would be preferred. For example, change "EM analysis indicated a significant increase of the number of MVBs..." to "EM analysis indicated a $\sim 70\%$ increase in the number of MVBs...."

8) Figure 3 legend: panels h and i are mistakenly referred to as c and d.

9) Figure 5G: where "RAF" is written on the X-axis, should it not be "BAF"?

10) Second paragraph of Results section "Overexpression of Rab27a rescues...." The authors state that Rab27a overexpression does not 'rescue' exosome secretion in control knockdown cells. Why would it 'rescue' if there was no defect in exosome secretion? In this case, I think the authors mean that Rab27a overexpression does not increase exosome expression relative to normal.

11) The authors should discuss Munc13-4 function in light of the recent paper by Thomas Martin and colleagues because Munc13-4 is an effector of Rab27.

12) KIBRA expression is tissue specific while exosomes are released by virtually all animal tissues. Are there KIBRA-like proteins expressed in other tissues?

Dear Editors and Reviewers:

We thank you for your valuable comments and suggestions concerning our manuscript entitled “KIBRA controls exosome secretion via regulating fusion between multivesicular bodies and lysosomes” (ID: NCOMMS-18-22098). We have carefully considered all the comments and suggestions, and wherever possible, incorporated into the revised manuscript. All major revisions are highlighted in yellow color. Of note, to fully address the comments from all the three reviewers, we have performed additional experiments to further explore the potential mechanisms of Rab27a degradation, and we found that Rab27a degraded mainly through ubiquitin-proteasome pathway rather than lysosomal pathway. Thus, we have changed the title of our manuscript to “**KIBRA controls exosome secretion via inhibiting the proteasomal degradation of Rab27a**”. Furthermore, the mass spectrometry proteomics data have been deposited to the ProteomeXchange Consortium (<http://proteomecentral.proteomexchange.org>) via the PRIDE database. The data is currently private, and can only be accessed with a single reviewer account (**Username:** reviewer27710@ebi.ac.uk, **Password:** NOHFPmhO). We provided point-by-point responses to the reviewers’ comments below (we copy the reviewer’s comments first, followed by our responses):

Reviewer #1:

1. Comment: *whether KIBRA regulates exosome secretion is a general phenomenon or tissue specific phenotype? If the authors would like to claim this is a general phenomenon, more than one cell line or tissues need to be test.*

Response: We thank the reviewer for the valuable suggestion. Given that KIBRA is predominately expressed in the kidney and the brain of mammals, we further examined this phenomenon in mouse podocyte cell line (MPC5) and in mouse kidney tissue. Our results indicated that knockdown of KIBRA in MPC5 cells as well as in HT22 cells led to a significant decrease in exosome secretion (Fig. 1 j-l). Further, we isolated and purified exosomes from the kidney of mice by differential centrifugation with sucrose density gradient. We found that exosomes isolated from kidney tissue of KIBRA-KO mice were significantly decreased compared with their WT counterparts (Fig. 3 a, d), which is consistent with the results in the brain. Taken together, these additional experiments support the view that KIBRA regulated exosome secretion is a general phenomenon rather than a tissue-specific phenotype. We have now incorporated these experiments into the revised manuscript (see methods on page 23 and pages 26–27, results on pages 6–7 and pages 10–12).

Fig. 1 j-l

Fig. 3 a, d

2. Comment: *The mechanism how loss of function KIBRA induced degradation of Rab27a was not studied.*

Response: We thank the reviewer for this suggestion. To explore the Rab27a degradation mechanisms, KIBRA-KD and Ctrl-KD cells were treated with the protein synthesis inhibitor (cycloheximide, CHX), the proteasome inhibitor (lactacystine, Lac) or the lysosome inhibitor (bafilomycin A1, Baf). Western blot analysis showed that Rab27a enormously degraded when treated with CHX for 12 h, and this degradation was restored by the proteasome inhibitor Lac but not by the lysosome inhibitor Baf (Fig. 5a, b). Furthermore, immunoprecipitation (IP) experiments showed that Rab27a was more easily ubiquitinated when KIBRA was depleted, and the Lac treatment dramatically increased the levels of ubiquitinated Rab27a (Fig. 5c, d). These results indicate that Rab27a was degraded mainly through the ubiquitin-proteasome pathway. Rab27a becomes stabilized through an interaction with KIBRA and is therefore free from being ubiquitinated. In contrast, depleting KIBRA leads to increased proteasomal

degradation of Rab27a, which in turn suppressed exosome secretion. We have now incorporated these experiments into the revised manuscript (see methods on pages 26 and 28, results on pages 15–17, and discussion on pages 20).

Fig. 5

3. Comment: Authors knockdown or overexpressed KIBRA in HT22 cells, however, the endogenous expression level of KIBRA was dramatically different in supplemental Figure 1A & B. Can authors explain this discrepancy?

Response: We thank the reviewer for the comment. Actually, HT22 cells are abundant in endogenous KIBRA. However, the endogenous expression level of KIBRA was dramatically different in our previous version of the manuscript. We think that the different gray-scale values of endogenous KIBRA could be due to differences in the exposure time. To avoid misunderstanding, we have replaced previous Supplementary Fig. 1B with an updated figure (Supplementary Fig. 2B). We have now reorganized the figures in the revised manuscript, in which the Supplementary Fig. 2A and B in the

revision corresponded to Supplementary Fig. 1A and B in the previous version.

Supplementary Fig. 2A, B

4. Comment: *Figure 2, a second marker other than CD63 needs to be included to confirm the authors' conclusion.*

Response: We thank the reviewer for the excellent suggestion, which will indeed help make our conclusions more convincing. To fully address this comment, we performed immunofluorescence analysis using lyso-bisphosphatidic acid (LBPA) as another MVB marker (Kobayashi et al. Nature1998;392:193-197). Consistent with immunofluorescence images of CD63, our results showed that the size and number of LBPA-positive MVBs in KIBRA-KD cells were significantly increased compared with control cells. We have now reported this additional experiment in the revision (see methods on page 23 and pages 26–27, results on pages 8–10).

Fig. 2g

5. Comment: *Figure 3, is it brain specific KIBRA knockout in Wwc1tm1.1Rlh mice? If not, can authors detect exosome secretion difference in the blood compare to wild type*

mice?

Response: We apologize for having not provided sufficient information about Wwc1tm1.1Rlh mice. According to the Jackson Laboratory web site (<https://www.jax.org/strain/024415>), KIBRA was completely knockout in all the tissues of Wwc1tm1.1Rlh mice. We have added this information to the “Materials and methods” section as follows (see page 25): “Wwc1tm1.1Rlh mice were purchased from the Jackson Laboratory (No. 024415; Bar Harbor, ME, USA). KIBRA was completely knocked out in all tissues of this strain.” In addition, exosomes were isolated from the same volume of serum from KIBRA-KO and WT mice following the sequential centrifugation steps. NTA analysis showed that the number of exosomes isolated from KIBRA-KO mice was only about 52% of that from the WT mice (Fig. 3e, f), suggesting that the exosome secretion in peripheral blood serum was impaired in KIBRA-KO mice. We now added this additional results to the revision (see pages 10–13).

Fig. 3e, f

6. Comment: *Although authors claimed the GFP-KIBRA punctae displayed a significant, but not complete, colocalization with the MVBs marker CD63. However, Figure 4A showed GFP-KIBRA exclusively localized with MVBs.*

Response: Thank you very much for your careful checks. In response to the reviewer’s previous comment (see comment no. 2), we have further explored the potential mechanisms of Rab27a degradation and found that Rab27a was degraded mainly through ubiquitin-proteasome pathway rather than lysosomal pathway. Thus, we have modified the text in the revised manuscript (see pages 15–17) and deleted this figure.

Reviewer #2:

1. Comment: *The work is interesting, relatively novel : roles for KIBRA in exosome or EV secretion, or in RAB27 protein stability have never been described, although other roles for KIBRA in intracellular trafficking and exocytosis have been published (ref 21 Traer et al 2007, and Yoshihama et al. 2011. Curr Biol 21:705, not quoted, slightly contradictory with this work).*

Response: We thank the reviewer for the generally positive and valuable comments. The reviewer kindly drew our attention to the work by Yoshihama and colleagues, which is indeed an excellent paper. We have briefly discussed the role for KIBRA in exocytosis and cited this paper in the revised manuscript (see page 3 and ref. 23). The work by Yoshihama and colleagues demonstrated that KIBRA regulated epithelial cell polarity by suppressing apical exocytosis. In our paper, KIBRA was found to improve to some extent secretion of EVs. This probably suggests that exocytosis and EV secretion may represent two different biological processes or mechanisms that are involved in membrane trafficking.

2. Comment: *The authors should start their article using the generic term « small EVs », before moving to the term exosomes, when they more specifically analyse MVBs. And they should also analyse the effects of KIBRA KO/KD on secretion of other types of EVs, such as larger EVs recovered by lower speed centrifugation.*

Response: We thank the reviewer for the constructive and valuable suggestions. We have changed the term “exosomes” into “small EVs” in the “Results” section on pages 5–7. We have also added brief explanations on this issue to the “Discussion” section on page 20. Furthermore, we have analysed the effect of KIBRA knockdown on secretion of larger EVs recovered by lower speed centrifugation ($2,000 \times g$ for 10 min = 2K pellet and $10,000 \times g$ for 30 min = 10K pellet). As expected, we found that KIBRA also decreased secretion of larger EVs (Supplementary Fig. 3), although the differences of 2K and 10K pellets were not as significant as ultracentrifuged pellets (small EVs). We now reported this experiment in the revision (see methods on pages 26-27, results on pages 5–7 and Supplementary Information).

Supplementary Fig. 3

3. Comment: *The model of Rab27a degradation by lysosomes through fusion of MVB with lysosomes induced in the absence of KIBRA, however, does not satisfy me. Most likely, degradation of Rab27a observed by the authors is due to another mechanism of degradation, maybe proteasome-mediated (the authors must evaluate the effect of proteasome inhibitors like lactacystine on Rab27a level in wt and kibra ko cells). Another possible mechanism would be by direct or indirect association between KIBRA and Rab27a, which would stabilize the latter: the authors should test whether Rab27a is co-precipitated with KIBRA. Alternatively, since this part of the article is not very important for the rest of the message, I would suggest to delete fig4, fig5f-g and fig7.*

Response: We really appreciate and thank the reviewer for the thoughtful and valuable comments. With regard to the mechanisms of Rab27a degradation, we totally agree with the reviewer that there are indeed several issues in the previous draft that need to be addressed. Accordingly, a series of additional experiments were performed to clarify the degradation mechanisms of Rab27a, which is the key point of our study:

Firstly, to explore the Rab27a degradation mechanisms, KIBRA-KD and Ctrl-KD cells were treated with the protein synthesis inhibitor (cycloheximide, CHX), the proteasome inhibitor (lactacystine, Lac) or the lysosome inhibitor (bafilomycin A1, Baf). Western blot analysis showed that Rab27a enormously degraded when being treated with CHX for 12 h, and this degradation was restored by the proteasome inhibitor Lac but not by

the lysosome inhibitor Baf (Fig. 5a, b). Furthermore, immunoprecipitation (IP) experiments showed that Rab27a was more easily ubiquitinated when KIBRA was depleted, and the Lac treatment dramatically increased the levels of ubiquitinated Rab27a (Fig. 5c, d). These results indicate that Rab27a was degraded mainly through the ubiquitin-proteasome pathway.

Secondly, cross IP and immunofluorescence co-localization analyses provided direct evidence supporting the Rab27a/KIBRA interactions (Fig. 5e-g). Rab27a becomes stabilized through an interaction with KIBRA and therefore, it is free from being ubiquitinated. In contrast, depleting KIBRA leads to increased proteasomal degradation of Rab27a and in turn suppressed exosome secretion. We have now carefully incorporated these experiments in the revision (see methods on page 26 and 28, results on pages 15–17, and discussion on page 20).

Fig. 5

4. Comment: Figure 1C showing the level of different EV-associated proteins in the isolated ultracentrifuged pellets from KIBRA-KD or -OE cells should also show the

levels of these proteins in the cell lysates to determine if the differences in secretion levels are due to impaired secretion of EVs or rather to decreased intracellular levels of the EV markers.

Response: The reviewer is absolutely right. We have now shown the levels of these proteins in the whole cell lysates (WCL) in the revised manuscript (see pages 5–7). Knockdown or overexpression of KIBRA in HT22 and MPC5 cells only influence the expression of Alix, Tsg101, CD63, and CD9 in ultracentrifuged pellets but not in the whole cell lysates (WCL) (Fig. 1b, h, and k).

Fig. 1

5. Comment: *EV isolation from brain tissue is performed in a way that separate eliminate fragments of cells, generated during tissue mincing and passing through needles. The authors quote ref 44 for this protocol, but they should be aware of more careful work performed on brain tissue, to isolate EVs and exosomes through gradient-based separation: Vella et al, 2017, J Extracell Vesicles 6, 1348885.*

Response: Thank you very much for your valuable suggestion. According to your suggestions, we isolated EVs and exosomes from the brain and kidney extracellular space of mice by differential centrifugation with sucrose density gradient following the protocols of Vella et al., J Extracell Vesicles 2017, 6, 1348885 (ref. 31) and Perez-Gonzalez et al., J Biol Chem 2012, 287, 43108-43115 (ref. 51). In the revised manuscript, we have modified the description of methods about “exosome isolation” (see pages 26–27). Results were shown in Fig. 3a-d.

Fig. 3a-d

6. Comment: *The anti-CD63 antibody should show a fuzzy band, due to high level of glycosylation, the sharp bands shown in WB are probably not specific.*

Response: Thank you very much for your careful checks. The anti-CD63 antibody we used in the previous manuscript was ab213092 (Abcam). In the revised manuscript, we have changed the CD63 antibody to ab217345 (Abcam) and all the western blot experiments of CD63 have been redone, and the manuscript has been accordingly updated (see methods on page 23, results on pages 7–12).

7. Comment: *The WB showing Rab27a bands in figure 6 should show position of both the endogenous Rab27a and the Rab27a-mcherry construct : it is not clear why Rab27a-Cherry overexpression should enhance expression of the endogenous protein (25kDa, fig 6a), nor if the transfected protein or upregulation of the endogenous one is responsible for recovery of the KIBRA-KO phenotype.*

Response: We thank the reviewer for kindly pointing out these errors. Firstly, Rab27a plasmid was indeed DsRed-tagged, but we mistakenly labeled it as “Rab27a-mCherry” in Fig. 6a and Fig. 6e, which have now been corrected in the revised manuscript (see page 18). Secondly, we had cut off the exogenous bands of Rab27a-DsRed unintentionally in the initial manuscript, which is really a big mistake. Uncropped western blot data of Rab27a in the previous manuscript (left panel) and in the revised manuscript (right panel) were shown as follows.

Fig. 6

8. Comment: *Suppl Table 1* does not seem right: column 1 = KO vs WT indicates values between 0.8 and 1.2 of FC for all proteins analysed, whereas column 2 = Log2FC gives much more variable values : what is column 1 showing ? the actual values of the 2 or 3 samples and/or the mean \pm sd would be more informative in this column.

Response: We again apologize for our carelessness. In the previous draft, column 2 should have been labeled as Log₂FC, but not Log2FC. We have corrected the column 1 from KO vs. WT FC to the mean \pm SD of WT and KO groups of mice (see Supplemental Table 1).

9. Comment: *In Figure 5a, explain the color code. Finally, these analyses should have been done (and would likely be more reproducible) in 3 independent samples of the CTRL-KO and KIBRA-KO cell lines.*

Response: We thank our reviewer for pointing out this issue. In the revised manuscript, we have made it clear that the up- and down-regulated proteins are indicated by red and blue hues, respectively. The color intensity indicates the expression levels of the proteins, as displayed (Fig. 4a).

Fig. 4a

We absolutely agree with the reviewer's suggestions that CTRL-KO and KIBRA-KO cell lines would likely be more reproducible and there is significant individual difference between KIBRA-WT and -KO mice. However, single cell clone of KIBRA-knockdown cells was not selected after infection with CRISPR-Cas9 lentivirus. Thus, KIBRA was not totally depleted in all cells of KIBRA-knockdown group. To minimize the individual difference between KIBRA-WT and -KO mice, KIBRA-/+ mouse was mated with KIBRA-/+ mouse, and littermate offspring of KIBRA-WT and -KO mice were selected to perform mass spectrometry (MS) analysis. To validate the results from MS analysis, we have compared the levels of Rab27a in the cortex and hippocampus of KIBRA-KO and -WT mice by western blot and immunofluorescence analyses. In line with the MS results, we found that KIBRA depletion led to a significant decrease of Rab27a protein levels. We have added these experiments to the revised manuscript (see pages 13–15).

10. Comment: *some typos and grammatical errors could be corrected throughout the manuscript.*

Response: We thank the reviewer for this comment. The leading authors have made all efforts to avoid any typos and grammatical errors in the revised manuscript. Furthermore, two of our senior co-authors (Y.W.; C.Q.) have made careful editorial revisions throughout the manuscript. We hope and believe that the revised manuscript could largely meet the language standards for publication in the Nature Communications.

Reviewer #3:

1. Comment: *Were KIBRA to have a 'switch' function that controls one or more targets involved in MVB-plasma membrane fusion, an active role in controlling exosome release would be more conceivable. Thus, as it stands, the authors' study reveals a requirement for KIBRA, but its role is not evident at this point. For example, does KIBRA directly bind to Rab27? It is puzzling that KIBRA also regulates dynein light chain, which is completely unrelated to exosome trafficking (as far as is known). Can KIBRA be a chaperone for Rab27 and other molecules?*

Response: We truly appreciate all the thoughtful comments and valuable suggestions. Through Cross immunoprecipitation (IP) and immunofluorescence co-localization analysis, we have provided solid evidence supporting the Rab27a/KIBRA direct interactions (Fig. 5e-g). In addition, IP experiments showed that Rab27a was more easily to be ubiquitinated when KIBRA was depleted and that lactacystin treatment significantly increased levels of ubiquitinated Rab27a (Fig. 5c, d). These results indicated that Rab27a becomes stabilized through an interaction with KIBRA and is therefore free from being ubiquitinated. In contrast, depleting KIBRA led to increased proteasomal degradation of Rab27a, which in turn suppressed exosome secretion (please refer to our responses to comment no. 2 of the reviewer #1 for more details). However, we agree with the reviewer that further research is still needed to investigate whether additional proteins may also be involved in this process.

Fig. 5

2. Comment: *There are numerous errors in syntax, grammar, and spelling; and the frequency of these errors increases as the manuscript progresses. More careful editing is recommended.*

Response: We thank the reviewer for kindly pointing out the language issue. To address this issue, the leading authors have made all efforts to avoid any typos and grammatical errors in the revised manuscript. Furthermore, two of our senior co-authors (Y.W.; C.Q.) have made careful editorial revisions throughout the manuscript.

3. Comment: *Results from NTA are presented in figure 1 without a description of this method of analysis or rationale for its use.*

Response: We thank the reviewer for this comment. In the revised manuscript, we have now added a brief description of NTA to the “Materials and methods” section on pages 27–28.

4. Comment: *Supplemental figure 1C, D: please clarify if protein or mRNA levels are measured.*

Response: We have actually performed western blot and qRT-PCR analysis to measure the protein and mRNA levels, respectively, of KIBRA-KD and -OE cells, and showed the results in Supplementary figure 2 (we renumbered the Supplemental figure 1C, D in the previous version as Supplementary fig.2 in the revision).

Supplementary figure 2

5. Comment: *Figure 1C would benefit from showing the abundance of each protein in total cellular extracts from control versus knockdown conditions.*

Response: We thank the reviewer for this very valuable suggestion, because protein levels of WCL determine whether the differences in secretion levels are due to impaired secretion of exosomes or decreased intracellular levels of the EV markers. We have shown the levels of these proteins in the whole cell lysates (WCL) in the revised manuscript (Fig. 1).

6. Comment: *The increased number of ILVs in MVBs of KIBRA-knockdown cells is intriguing. Do the authors have an idea of why this might be the case?*

Response: Ostrowski and colleagues reported that Rab27a silencing could affect exosome secretion through reducing MVB docking to the plasma membrane. In the absence of Rab27a, the size of MVBs was increased (Ostrowski M, et al., Nature Cell Biology 2010;12:19-30). In our study, we showed that KIBRA regulated exosome secretion through stabilizing Rab27a and the phenotypes of MVBs induced by KIBRA depletion are similar to that induced by silencing Rab27a. Therefore, KIBRA and Rab27a may share similar mechanisms in regulating exosome secretion. We have now added a brief discussion on this (see pages 21–22 and ref. 10).

7. Comment: *Figure 2D: please clarify if '20 cell profiles' refers to profiles of 20 different cells or includes serial sections of the same cells.*

Response: In fact, all the profiles used for quantification were obtained from different cells. In the revised manuscript, we have replaced “20 cell profiles” with “20 profiles of different cells” (see page 9). We thank the reviewer for pointing out this vague wording.

8. Comment: *In general, the term 'significant' is liberally used to describe differences in various parameters when comparing cells/mice that have normal versus alter KIBRA expression. More accurate descriptions would be preferred.*

Response: Indeed, the term “significant” is most frequently used in terms of statistical difference, but significant difference (or increase or decrease) does not reflect the extent of the differences. We agree with the reviewer that accurate description will help avoid misunderstanding. Whenever appropriate, we have replaced the term “significant” with

more accurate and concrete descriptions. For example, we have changed “EM analysis indicated a significant increase of the number of MVBs and ILVs per cell in KIBRA-KO mice compared with their WT littermates, and ILVs per MVB were increased apparently as well.” to “The results showed a ~60% increase in the number of MVBs per cell in KIBRA-KO mice compared with their WT littermates. Meanwhile, the number of ILVs per cell and the number of ILVs per MVB increased by ~120% and ~40%, respectively.” (see page 11).

9. Comment: *Figure 3 legend: panels h and i are mistakenly referred to as c and d.*

Response: We thank the reviewer for kindly pointing out this error, and we have now amended it in the revised manuscript.

10. Comment: *Figure 5G: where “RAF” is written on the X-axis, should it not be “BAF”?*

Response: The reviewer is right. In our revised manuscript, however, this figure has been deleted, and we have carefully checked all the figures in the manuscript to avoid similar mistakes.

11. Comment: *Second paragraph of Results section “Overexpression of Rab27a rescues...” The authors state that Rab27a overexpression does not ‘rescue’ exosome secretion in control knockdown cells. Why would it ‘rescue’ if there was no defect in exosome secretion? In this case, I think the authors mean that Rab27a overexpression does not increase exosome expression relative to normal.*

Response: Yes, the reviewer is correct. We have now modified this sentence as “However, exosome secretion did not increase in Ctrl-KD cells, even though Rab27a was overexpressed.” (see page 17).

12. Comment: *The authors should discuss Munc13-4 function in light of the recent paper by Thomas Martin and colleagues because Munc13-4 is an effector of Rab27.*

Response: We thank the reviewer for kindly drawing our attention to the work by Martin and colleagues. It is indeed an excellent paper. We have now briefly discussed the Munc13-4 function and cited the paper by Martin and colleagues (see page 21 and ref. 39) in the revised manuscript.

13. Comment: *KIBRA expression is tissue specific while exosomes are released by virtually all animal tissues. Are there KIBRA-like proteins expressed in other tissues?*

Response: The reviewer’s comment is of great importance and deserves further discussion in our manuscript. Thus, we have added the brief comments to the revised manuscript (see pages 21–22).

REVIEWERS' COMMENTS:

Reviewer #1 (Remarks to the Author):

The authors have made great efforts to address the points raised in the previous review. I think the manuscript is ready to be published.

Reviewer #2 (Remarks to the Author):

In this revised article by Song et al, the authors have performed a large array of new experiments to answer my previous comments, which led them to amend their interpretations and model, which is now satisfyingly demonstrated

The authors should consider providing the following remaining minor modifications:
figures 4-6 still contain inappropriate bar graphs instead of dot plot representations of individual biological replicates: please convert to dot plots (it is also acceptable to retain a bar graph presentation if individual dots of biological replicates are positioned on the bars)

Reviewer #4 (Remarks to the Author):

The authors adequately addressed my comments with new experiments or clarifications/discussions. I consider these responses satisfactory.

REVIEWERS' COMMENTS:

Reviewer #1 (Remarks to the Author):

The authors have made great efforts to address the points raised in the previous review. I think the manuscript is ready to be published.

Reviewer #2 (Remarks to the Author):

In this revised article by Song et al, the authors have performed a large array of new experiments to answer my previous comments, which led them to amend their interpretations and model, which is now satisfyingly demonstrated

The authors should consider providing the following remaining minor modifications: figures 4-6 still contain inappropriate bar graphs instead of dot plot representations of individual biological replicates: please convert to dot plots (it is also acceptable to retain a bar graph presentation if individual dots of biological replicates are positioned on the bars)

Reviewer #3 (Remarks to the Author):

The authors adequately addressed my comments with new experiments or clarifications/discussions. I consider these responses satisfactory.

Response: We appreciate the reviewers for their positive comments. According to the reviewer's suggestion, we have converted the bar graphs to dot plots in Figures 4-6 in the revised manuscript.